# MODELLING LONG RANGE DEPENDENCIES IN $N$D: FROM TASK-SPECIFIC TO A GENERAL PURPOSE CNN

**David M. Knigge**[*,1], **David W. Romero**[*,2], **Albert Gu**[3], **Efstratios Gavves**[1],
**Erik J. Bekkers**[1], **Jakub M. Tomczak**[2], **Mark Hoogendoorn**[2], **Jan-Jakob Sonke**[4]
[1] University of Amsterdam    [2] Vrije Universiteit Amsterdam    [3] Stanford University
[4] Netherlands Cancer Institute
d.m.knigge@uva.nl, d.w.romeroguzman@vu.nl

## ABSTRACT

Performant Convolutional Neural Network (CNN) architectures must be tailored to specific tasks in order to consider the length, resolution, and dimensionality of the input data. In this work, we tackle the need for problem-specific CNN architectures. We present the *Continuous Convolutional Neural Network* (CCNN): a single CNN able to process data of arbitrary resolution, dimensionality and length without any structural changes. Its key component are its *continuous convolutional kernels* which model long-range dependencies at every layer, and thus remove the need of current CNN architectures for task-dependent downsampling and depths. We showcase the generality of our method by using the *same architecture* for tasks on sequential (1D), visual (2D) and point-cloud (3D) data. Our CCNN matches and often outperforms the current state-of-the-art across all tasks considered.[1]

## 1 INTRODUCTION

The vast popularity of Convolutional Neural Networks (LeCun et al., 1998) (CNNs) is a result of their high performance and efficiency, which has led them to achieve state-of-the-art in applications across sequential (Abdel-Hamid et al., 2014; Van Den Oord et al., 2016), visual (Krizhevsky et al., 2012; Simonyan & Zisserman, 2014) and high-dimensional data (Schütt et al., 2017; Wu et al., 2019). Nevertheless, a major limitation of CNNs –and other neural networks– is that their architectures must be tailored to particular applications in order to consider the length, resolution and dimensionality of the input data. This has led to a plethora of task-specific architectures (Oord et al., 2016; Bai et al., 2018; Simonyan & Zisserman, 2014; Szegedy et al., 2015; Ronneberger et al., 2015; He et al., 2016; Qi et al., 2017; Wu et al., 2019) which (*i*) hampers the selection of the most appropriate architecture for a particular task, and (*ii*) obscures the transfer and generalization of insights across applications. In this work, we tackle the need for problem-specific CNN architectures and propose a generic CNN architecture that can be used independent of the length, resolution and dimensionality of the data.

**CNN architectures are data dependent.** Current CNN architectures are task-specific because they are tied to the *length*, *resolution*, and *dimensionality* of the input. The *length* of the data varies from task to task, e.g. audio fragments may span milliseconds to minutes. This requires carefully chosen

---

[*]Equal contribution.

[1]Our code is publicly available at github.com/david-knigge/ccnn.

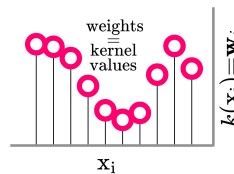
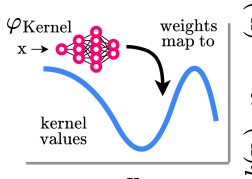

Figure 1: Discrete and continuous convolutional kernels. Discrete convolutional kernels assign a weight $\mathbf{w}_i$ out of a discrete set of weights $\mathbf{W}$ to a relative offset $\mathbf{x} - \tilde{\mathbf{x}}$. This ties the kernel to the length, resolution and dimensionality of the input, limiting the general applicability of the CNN architectures. Instead, our *Continuous Convolutional Neural Network* parameterizes kernel values as a continuous function $\varphi_{\text{Kernel}}$ over the input domain $\mathbb{R}^d$, which decouples it from data characteristics.

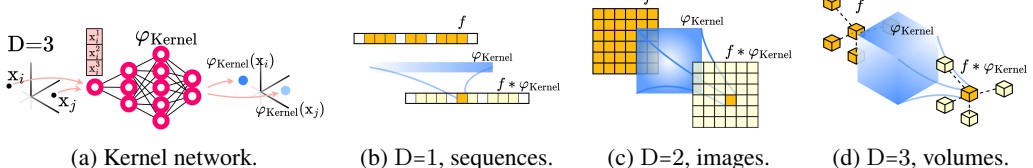

(a) Kernel network.  (b) D=1, sequences.  (c) D=2, images.  (d) D=3, volumes.

Figure 2: Continuous convolutional kernels: the key to a unified CNN architecture. The continuous parameterization of convolutional kernels used in this work consists of a small kernel network $\varphi_{\text{kernel}}$ that receives coordinates as input and outputs the value of the convolutional kernel at that position (2a). By changing the dimensionality of the coordinates $\mathbf{x}_i$, the same kernel network can render convolutional kernels for sequential (2b), visual (2c), and higher dimensional data (2d).

strides and pooling to capture relevant dependencies across the entire input (Van Den Oord et al., 2016; Lee et al., 2017). In addition, physical signals, e.g., audio, images, are continuous in nature. As such, their semantic meaning is independent of the *resolution* at which they are sampled, e.g., the same audio may be expressed at different resolutions. Nevertheless, current CNN architectures are resolution-bound, and thus different resolutions require different CNNs. These limitations aggravate when considering *multi-dimensional* data. Each input dimension can be defined at different lengths and resolutions, e.g., video, rectangular images, and each data modality brings its own conventions for each of these properties, e.g., the resolution of a second of audio (16kHz) (Warden, 2018) strongly contrasts with that of images ($32 \times 32$) (Krizhevsky et al., 2009).

**Towards a unified CNN architecture.** As discussed in Sec. 3, the core component that makes CNNs data-dependent are their *discrete convolutional kernels*. Convolutional kernels are implemented via a one-to-one mapping between kernel values and model parameters (Fig. 1 left), which (*i*) binds them to the input resolution and length, and (*ii*) makes them ill suited to model long-range dependencies. The latter results from the large number of parameters needed to construct large convolutional kernels. This is why standard CNNs favour using local kernels in combination with task-dependent depths and pooling layers to model long-range dependencies, at the cost of making them task-dependent.

**The need for a continuous parameterization.** To overcome task-dependent architectures, it is crucial to define a kernel parameterization that decouples parameter count from kernel size. Following Schütt et al. (2017); Romero et al. (2022b), we use a small neural network to define a continuous mapping from positions to the value of the kernel at those positions. The resulting *Continuous Convolutional Kernels* (Fig. 2), allow for the construction of convolutional kernels of arbitrary size in a parameter efficient manner. Consequently, the same convolutional layers –and thus the same CNN– can be used regardless of the input length, resolution and dimensionality. We leverage this formulation to construct the *Continuous Convolutional Neural Network* (CCNN): a single CNN architecture that can be applied regardless of the input length, resolution and dimensionality.

**Empirical results.** To showcase the proposed CCNN, we deploy the same CCNN for several tasks on sequential (1D), visual (2D) and point-cloud (3D) data. Our CCNN matches and often outperforms the current state-of-the-art across all tasks considered. Importantly, the continuous parameterization of our CCNN allows it to handle irregularly sampled data natively. As a result, the CCNN is not restricted to grid data, e.g., 3D voxels, and can be used on point-clouds directly.

**Contributions:**
- We propose the *Continuous Convolutional Neural Network*: a general purpose CNN architecture able to process data of arbitrary resolution, dimensionality and length without structural changes.
- We study the layers of CNNs, and demonstrate that the ability to model long-term dependencies on $N$D without the need of input dependent downsampling and depth values is *necessary and sufficient* for the construction of a general purpose CNN architecture.
- In order to model long-term dependencies on $N$D without input dependent downsampling and depth values, we utilize and improve the Continuous Kernel Convolutions of Romero et al. (2022b). Our proposed improvements allow the proposed Continuous CNN to achieve good empirical results on the tasks considered in 1D, 2D and 3D without structural changes.

## 2 RELATED WORK

An extended section with extended comparisons to related works is provided in Appx. A.

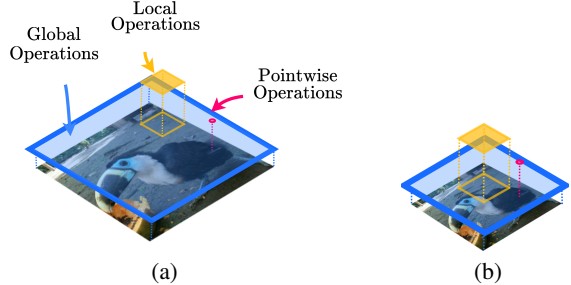

(a)                    (b)

Figure 3: Operation types: global, pointwise and local. Local operations are resolution dependent. Transferring a local operation from (3a) to a lower resolution (3b) leads to an increased receptive field.

**General purpose architectures.** To the best of our knowledge, the only existing method aiming for a general purpose architecture is the Perceiver (Jaegle et al., 2021), which uses a Transformer to lift restrictions regarding data characteristics and modalities. However, (*i*) it must map inputs –regardless of their size– to a small latent representation to reduce the quadratic complexity of self-attention, (*ii*) decouples the depth of the network from its parameter count via recurrence, which requires tuning the number of unrolling steps per task, and (*iii*) must use absolute positional encodings to encode the data structure, which break the translation equivariance of the self-attention operation (Romero & Cordonnier, 2020). In contrast, CCNNs provide a general purpose convolutional method that: (*i*) scales much more favorably than self-attention, (*ii*) does not require a constant small latent representation, (*iii*) does not require task-dependent depths, and (*iv*) preserves translation equivariance.

**Long range dependencies in $N$D.** Based on our analysis (Sec. 3), any architecture able to model long range dependencies in $N$D without the need of input-dependent pooling or depth could be used as a general purpose architecture. To our best knowledge, the only existing convolutional methods able to construct global convolutional kernels are CKConvs (Romero et al., 2022b;a) and state-spaces (Gu et al., 2021; 2022). However, state-spaces rely on complex dynamical systems that are not easily defined in $N$D –aside from the combination of 1D systems, equivalent to representing $N$D kernels as combinations of $N$ independent 1D kernels– Nguyen et al. (2022). Consequently, we select CKConvs as the building block of our approach given their advantages in terms of expressivity and simplicity.

**Convolutional kernels as neural networks.** Modelling convolution kernels with small neural networks that map kernel positions to kernel values showed promising results for small kernels (Jia et al., 2016; Schütt et al., 2017; Wu et al., 2019). Subsequently, Romero et al. (2022b) realized that this parameterization decouples the size of the convolutional kernel from the number of parameters required to construct it, and thus can be used to construct arbitrarily large convolutional kernels in a parameter efficient manner. In this work, we show that this parameterization allows for the construction of a single CNN that can be used regardless of the input length, resolution and dimensionality.

Romero et al. (2022b) realized that the piece-wise MLPs used so far to parameterize convolutional kernels were unable to model complex long range dependencies due to their spectral bias (Tancik et al., 2020), and showed that implicit neural representations –specifically SIRENs (Sitzmann et al., 2020)– could be used to solve the issue. Subsequently, Romero et al. (2022a) parameterized their convolutional kernels with Multiplicative Anisotropic Gabor Nets (MAGNets), which provided them control over frequencies admitted in the convolutional kernel, thus preventing aliasing and aiding generalization across resolutions. In our work, we use FlexConvs parameterized by MAGNets. However, we observe that neural networks used to parameterize convolutional kernels are not correctly initialized for that purpose, and propose an initialization that solves the issue.

## 3    FROM DATA-DEPENDENT TO DATA-INDEPENDENT CNN ARCHITECTURES

In this section, we study the components of CNN architectures and pinpoint the changes required in order to construct a CNN architecture independent of input lengths, resolutions and dimensionalities.

### 3.1    POINTWISE OPERATIONS: LINEAR LAYERS, DROPOUT, POINTWISE NONLINEARITIES AND RESIDUAL CONNECTIONS

Pointwise operations are operations applied to each spatial element of the input separately (Fig. 3), e.g., pointwise linear layers, dropout and pointwise nonlinearities. As such pointwise operations do not depend on the input shape and model the same function regardless of input length, resolution and dimensionality. Learnable parameters of pointwise operations, e.g., in pointwise linear layers and

Parametric ReLU (He et al., 2015), are shared over the spatial domain; the same set of parameters is applied to inputs of any spatial shape.

**Conclusion.** Based on the previous observations, we can conclude that pointwise operations can be used without changes in order to construct a general-purpose CNN architecture.

### 3.2 GLOBAL OPERATIONS: NORMALIZATION LAYERS AND GLOBAL POOLING

Global operations aggregate all spatial elements of the input for their processing, e.g., normalization layers, global pooling. As such, common global operations do not depend on the specific shape of the input signal and their effect is equivalent regardless of the input length, resolution and dimensionality. It is important to note, however, that common global CNN operations only define learnable parameters along the channel axes e.g., the scale and mean parameters of a normalization layer have shape $\mathbf{w} \in \mathbb{R}^{N_{in}}$. Nevertheless, if one were to define a global operation for which spatial positions are also assigned weights, the shape of the parameters would be dependent on the input length, resolution and dimensionalty, and thus the previous statement would no longer hold.

**Conclusion.** The previous observations indicate that global operations that only define channel-wise learnable parameters can be used without changes in order to construct a unified CNN architecture. Luckily, this is the case for all common global operations used in CNN architectures.[2]

### 3.3 LOCAL OPERATIONS: CONVOLUTIONAL LAYERS AND SUBSAMPLING

Local operations are operations that rely on spatial portions of the input and are applied across its spatial dimensions, e.g., convolution, subsampling. In practice, the neighborhoods on which these operations are applied are hyperparameters, e.g., $3 \times 3$ kernels or pooling over $2 \times 2$ regions, and are selected based on the input size. Unfortunately, if the resolution of the input changes then the portion of the input that the operation considers changes and the effect of the operation changes (Fig. 3). Similarly, if the length of the input changes, then larger neighborhoods are required in order to model dependencies across the input. These effects exacerbate if one considers multi-dimensional inputs, as different dimensions might require different neighborhood sizes. Consequently, we can say that local operations *depend on the resolution, length and dimensionality of the input*.

A possible solution would be to adjust the size of the operations proportional to resolution and length changes. Unfortunately, if the local operation defines learnable parameters over its spatial dimensions –as in (discrete) convolutional kernels (Sec. 3.3.1)–, then increasing the size of the convolutional kernel is tied to a proportional increase in the number of parameters required to construct it. This, in turn changes the learning dynamics of the network and easily becomes prohibitive.

**The need for a continuous parameterization.** A better solution results from using a parameterization in which the number of parameters is *independent* from the size of the kernel. By doing so, the same parameterization can be used independently from the input length, resolution and dimensionality.

### 3.3.1 FROM DATA-DEPENDENT TO DATA-INDEPENDENT CONVOLUTIONAL LAYERS

Conventional CNNs implement a discrete version of the convolution operation defined as:

$$(f * k)^o(\mathbf{x}) = \int_{\mathbb{R}^D} f(\mathbf{x} - \tilde{\mathbf{x}})k^o(\tilde{\mathbf{x}})d\tilde{\mathbf{x}}, \quad o \in [1, ..., N_{out}), \tag{1}$$

where $f : \mathbb{R}^D \to \mathbb{R}^{N_{in}}$ is a D–dimensional input signal with $N_{in}$ channels, and $k : \mathbb{R}^D \to \mathbb{R}^{N_{in} \times N_{out}}$ is a set of $N_{out}$ convolutional kernels. For the types of data CNNs are commonly used for, e.g. images, the signal $f$ is generally sampled on a discrete grid of equidistant points, and thus it can be described as a function $f : \mathbb{Z}^D \to \mathbb{R}^{N_{in}}$. In practice, the signal $f$ is non-zero only on a finite subset of the grid $\Omega(f) \subset \mathbb{Z}^D$ with limits given by the range of sampling, e.g., height and width of an image. Accordingly, the convolutional kernel $k$ is defined over the same grid of coordinates, of which generally a subset $\Omega(k) \subset \Omega(f)$ maps to nonzero values. Since both $\Omega(k)$ and $\Omega(f)$ are discrete and finite, the convolution can be implemented as the inner product of function and kernel values at each position:

$$(f * k)^o(\mathbf{x}) = \sum_{\tilde{\mathbf{x}} \in \Omega(k)} f(\mathbf{x} - \tilde{\mathbf{x}})k^o(\tilde{\mathbf{x}}), \quad \mathbf{x} \in \Omega(f). \tag{2}$$

**Discrete convolutional kernels tie convolutional layers to data characteristics.** Conventionally, the convolutional kernel $k$ is implemented through a discrete set of randomly initialized weights $\mathbf{W}$, of which each entry $\mathbf{w}_i \in \mathbb{R}^{N_{in} \times N_{out}}$ corresponds to a point $\mathbf{x}_i \in \Omega(k)$. Consequently, an increased

---

[2]Global discrete convolutional kernels could be interpreted as global operations for which parameters along the spatial dimensions of the input are defined. Consequently, these require special treatment (Sec. 3.3).

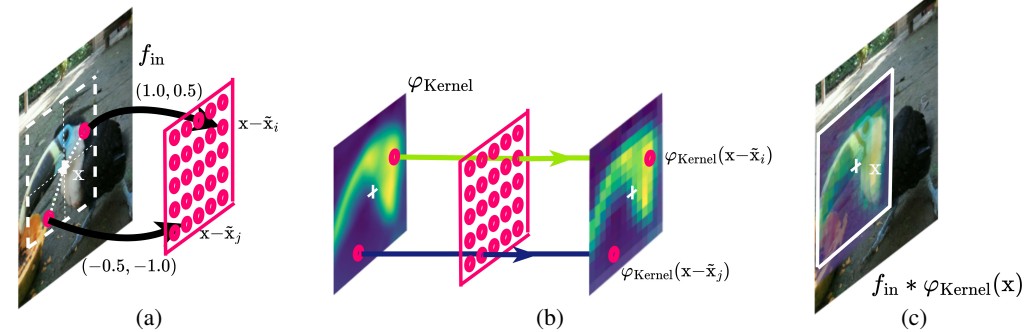

Figure 4: Applying a Continuous Kernel Convolution. Given a pixel-position in the input image $\mathbf{x} \in \Omega(f)$, we obtain relative offsets to surrounding pixels $\{\mathbf{x} - \tilde{\mathbf{x}}\}_{\tilde{\mathbf{x}} \in \Omega(f)}$ (4a). Next, we pass each relative position to the kernel generator network in order to generate the kernel: $\{\varphi_{\text{Kernel}}(\mathbf{x} - \tilde{\mathbf{x}})\}_{\tilde{\mathbf{x}} \in \Omega(f)}$ (4b). Finally, we apply the convolution between the generated convolutional kernel and the input (4c).

kernel size is reflected in a larger set $\Omega(k)$, and thus in a correspondingly larger weight matrix $\mathbf{W}$. This directly ties a model's parameter count to its kernel size.

In order to model the long-range dependencies needed to extract high-level features in a parameter-efficient way, we must then resort to pooling operations and the stacking of layers that implicitly increase the receptive field of the kernel. This in turn, makes CNNs effective only on inputs of a certain size. For example, if we apply a network created to model long-range dependencies over images of size $256 \times 256$ to images of size $32 \times 32$, then intermediary pooling layers would make the spatial extent of the feature maps collapse long before all convolutional layers are applied. Similarly, if we use a network designed to model long-range dependencies over images of size $32 \times 32$ to images of size $256 \times 256$, then the network will not be able to model long-range dependencies in the input.

Additionally, the definition of the kernel $k$ through a discrete set of weights $\mathbf{W}$ ties the model to a given input resolution. Yet, for many applications, the input $f$ is a discretization of an underlying continuous function. Therefore, we would like our model to provide the same responses regardless of the resolution at which $f$ is provided. As discrete convolutional kernels live in a discrete domain, they cannot be easily represented at other resolutions. In fact, one can show that discrete CNNs do not generalize to unseen resolutions Romero et al. (2022a); Nguyen et al. (2022). Consequently, it is not possible to reliably apply trained discrete CNNs across resolutions.

In addition, note that in Eq. 2, the same discrete convolutional kernel $k$ can be used at every location $\mathbf{x} \in \Omega(f)$ only because the input signal $f$ is defined over an equidistant grid, and the values of the discrete kernel $k(\tilde{\mathbf{x}})$ align with the features $f(\mathbf{x}), \forall \mathbf{x} \in \Omega(f)$. This is not the case for irregular data. Consequently, discrete kernels are ill-suited to handle irregular data. With weights fixed to relative positions, an infinite number of weights would be needed to cover any continuous domain.

These limitations suggest a better approach to model convolutional kernels: *using the model weights to parameterize $k$ as a continuous function over the data domain $\mathbb{R}^d$.*

**A data independent parameterization.** To obtain a formulation for a CNN applicable to arbitrary resolutions and sizes, we require a parameterization for convolutional layers that is invariant to the set $\Omega(f)$ over which $f$ is sampled. In other words, we must find a parameterization with which the kernel can be modelled over the underlying continuous domain of the input signal, i.e., $\mathbb{R}^d$. Moreover, to avoid models with different parameter count for different resolutions, it is necessary that the parameterization of the kernel decouples its parameter count from the size of the kernel.

Such a parameterization is provided by *Continuous Kernel Convolutions* (CKConvs) (Romero et al., 2022b;a). CKConvs provide a continuous parameterization for convolutional kernels by using a small neural network $\varphi_{\text{Kernel}} : \mathbb{R}^D \to \mathbb{R}^{N_{\text{out}} \times N_{\text{in}}}$ as a kernel generator network. This network maps coordinates in the domain of the kernel $\mathbf{x}_i \in \mathbb{R}^D$ to the values of the convolutional kernel at that position: $k(\mathbf{x}) \in \mathbb{R}^{N_{\text{out}} \times N_{\text{in}}}$ (Fig. 1). A Continuous Kernel Convolution is illustrated in Fig. 4.

Since the parameter count of the kernel generator network is independent from the number of points in the neighborhood that determines the size of the kernel, CKConvs allow for construction of arbitrarily large kernels without increasing the parameter count of the layer. Romero et al. (2022b) shows that

large kernels allow CNNs to model long range spatial dependencies at every layer, thus removing the need for downsampling and stacking of layers to increase receptive fields. This in turn allows us to build an architecture which does not contain resolution-, dimensionality-, and size-dependent layers.

**Conclusion.** The previous observations indicate that local operations equipped with existing parameterizations are tied to the length, resolution and dimensionality of the input. Nevertheless, this limitation can be lifted if an alternative parameterization is used that detaches the neighborhood of action of the operation from the length, resolution and dimensionality of the input.

## 4   A GENERAL PURPOSE CNN ARCHITECTURE

**Starting point.** In principle, the CNN architectures with CKConv (Romero et al., 2022b) and FlexConv (Romero et al., 2022a) introduced previously fulfill the requirements posited in Sec. 3. Nevertheless, as depicted in these papers, these architectures still must be tailored to specific applications and domains in order to perform well, e.g., TCN (Bai et al., 2018) and ResNet (He et al., 2016) backbones for sequential and visual tasks, respectively in Romero et al. (2022a).

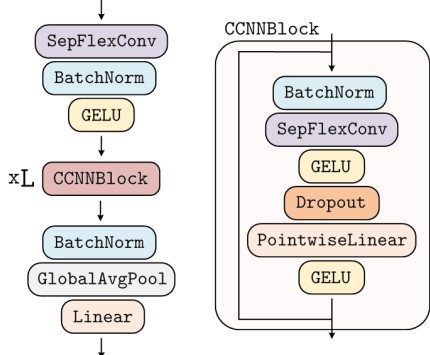

Figure 5: The CCNN architecture.

In order to construct a single performant general purpose architecture that works well across all tasks considered, we start from a FlexNet (Romero et al., 2022a), and propose several structural changes (Sec. 4.1). The resulting CCNN architecture is shown in Fig. 5.

### 4.1   MODIFICATIONS AND IMPROVEMENTS

**Proper initialization of the kernel generator network** $\varphi_{\text{Kernel}}$**.** First, we analize the kernel generator network $\varphi_{\text{Kernel}}$, and observe that it is not initialized properly in previous works (Schütt et al., 2017; Wu et al., 2019; Romero et al., 2022b;a) for the purpose of parameterizing convolutional kernels.

Recall that, in order to ensure training stability it is desirable to retain a constant (unitary) variance throughout the activations of a neural network (Glorot & Bengio, 2010). Hence, the discrete weights $\mathbf{W}$ conventionally used to construct a convolutional kernel are initialized to have variance inversely proportional to the number of elements over which the convolution is computed, i.e., the number of pixels $|\Omega(k)|$ times the number of channels $\text{N}_{\text{in}}$. For instance, He initialization (He et al., 2015) initializes $\mathbf{W}$ s.t. $\text{Var}(\mathbf{W}){=}g^2/(\text{N}_{\text{in}}|\Omega(k)|)$, with a gain $g$ that depends on the nonlinearity used.

In related works, the networks used to parameterize convolutional kernels are themselves initialized to *preserve a constant (unitary) variance throughout the network*. Consequently, when used as a kernel generator network, standard initializations lead the generated kernel to have unitary variance, i.e., $\text{Var}(k){=}1$. This in turn, make CNNs using neural networks to parameterize their convolutional kernels expe-

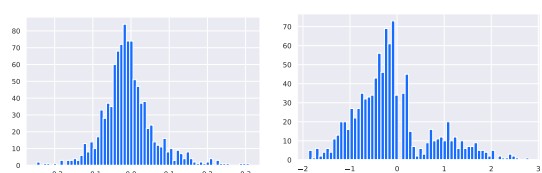

Figure 6: Histogram of the output of a CCNN with and without the proposed kernel initialization.

rience a layer-wise growth in the variance of the feature maps proportional to $\text{N}_{\text{in}} \cdot |\Omega(\text{k})|$. This growth is of particular importance for kernel generator networks that generate large convolutional kernels, i.e., with large $|\Omega(k)|$. For instance, we observe that the logits of CNNs with CKConvs (Romero et al., 2022b) and FlexConvs (Romero et al., 2022a) lie in the order of $1e^{19}$ upon initialization: an undersirable property that might lead to unstable training and a need for low learning rates.

To address this issue, we must ensure that the variance at the output of the kernel generator network is inversely proportional to $\text{N}_{\text{in}} \cdot |\Omega(\text{k})|$. Inspired by Chang et al. (2020), we therefore re-weight the last linear layer of the kernel generator network $\varphi_{\text{Kernel}}$ by $g^2/\sqrt{\text{N}_{\text{in}} \cdot |\Omega(\text{k})|}$. With this modification, we observe that the variance of the generated convolutional kernels satisfies the desired constrains, and consequently, the logits of our CCNN show unitary variance upon initialization (Fig. 6).

**Depthwise Separable Continuous Convolutions.** Separable convolutions have long been used to improve the parameter and computational efficiency of CNNs (Rigamonti et al., 2013; Sifre & Mallat, 2014). Recent architectures have leveraged separability, and found CNNs with separable kernels to

perform better than CNNs with conventional convolutions (Knigge et al., 2021; Liu et al., 2022). This phenomenon results from the separation of spatial and channel dimensions, which allows for wider networks without additional computational and parameter complexity.

Based on these observations, we construct a depth-wise separable version of FlexConv (Romero et al., 2022a), in which a channel-wise convolution is computed with a kernel generated by a kernel generator network $\varphi_{\text{Kernel}} : \mathbb{R}^D \to \mathbb{R}^{N_{\text{in}}}$, followed by a pointwise linear layer from $N_{\text{in}}$ to $N_{\text{out}}$ dimensions. Separable FlexConvs allow for the construction of a much wider CCNN –from 30 to 140 hidden channels– with the same parameter and computation complexity.

**An improved residual block.** Residual connections (He et al., 2016) provide training stability and improved performance. A residual block $\mathcal{R}(f)=\psi(f) + f$ is defined as the sum between the input and a so-called *residual connection* $\psi$ composed of multiple layers: convolutional, normalization, etc.

Recent studies found improvements over the residual block of He et al. (2016) by changing the nonlinearities as well as the position and type of normalization layers within the blocks (Xiong et al., 2020; Liu et al., 2022). Based on these advances and empirical evidence, we modify the residual blocks of Romero et al. (2022a) with residual blocks similar to those of S4 (Gu et al., 2022) complemented with a nonlinearity at the end of the block. A comparison of our residual block and those in Gu et al. (2022) and Romero et al. (2022a) is given in Fig. 7.

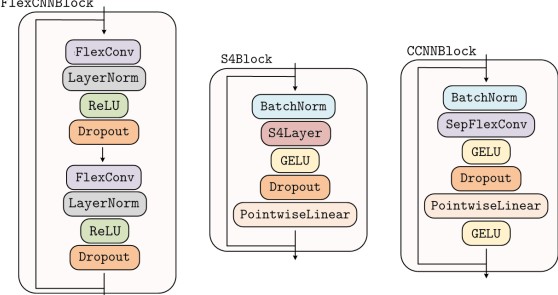

Figure 7: Residual blocks used in FlexCCNNs (Romero et al., 2022a), S4 (Gu et al., 2022) and CCNNs (ours).

$L_2$ **regularization of continuous kernels.** Weight decay penalizes the $L_2$ norm of learned kernel values $k$ by adding the norm of the weights $\mathbf{W}$ as an additional loss term (Krogh & Hertz, 1991). Since continuous kernels are not directly parameterized by weights $\mathbf{W}$, applying $L_2$ regularization on the parameters of $\varphi_{\text{Kernel}}$ directly would not have the intended effect. Consequently, in order to produce the same effect, we amend the $L_2$ regularization term to penalize the *generated convolution kernel* instead. Given $\varphi_{\text{Kernel}}^{:,l}$ the set of generated kernel for all channels at a layer $l$, $\mathcal{L}_{\text{obj}}$ the objective loss function, and $\lambda$ the regularizing parameter, we define a regularized loss as:

$$\mathcal{L} = \mathcal{L}_{\text{obj}} + \mathcal{L}_{\text{reg}} = \mathcal{L}_{\text{obj}} + \lambda \cdot \frac{1}{2} \sum_{l=1}^{L} \|\varphi_{\text{Kernel}}^{:,l}\|^2. \tag{3}$$

## 5 EXPERIMENTS

We aim to define an architecture that can be applied regardless of the specific data characteristics. To this end, we construct a CCNN and validate it on sequential (1D), visual (2D) and point-cloud (3D) datasets –see Appx. B for a detailed description of each dataset–. We show that the *same* CCNN obtains state-of-the-art results on several sequential tasks, competitive performance on image tasks, and surpasses the Perceiver on point-cloud processing. In addition, we show that learned CCNNs generalize from regular to irregular data by transferring a trained CCNN from voxels to point-clouds.

**Architecture specifications.** To study the scalability of our method, we create two CCNNs with differing numbers of hidden layers sizes: CCNN$_{4,140}$ (4 blocks, 140 channels, 200K params) and CCNN$_{6,380}$ (6 blocks, 380 channels, 2M params). The kernel network $\varphi_{\text{Kernel}}$ is chosen to be a 3-layer MAGNet (Romero et al., 2022a), with 32 hidden units for CCNN$_{4,140}$, and 64 hidden units for the larger CCNN$_{6,380}$. In each task, the input dimension of the kernel network $\varphi_{\text{Kernel}}$ corresponds to the dimensionality of the data –1 for sequences, 2 for images, and 3 for point-clouds–. Additional details regarding hyperparameters, training regimes and experimental settings are given in Appx. C. An empirical assessment of the computational efficiency of our architectures is given in Appx D.1.

**CCNNs on sequential datasets.** In 1D we consider Sequential MNIST, Permuted MNIST (LeCun et al., 1998), Sequential CIFAR10 (Krizhevsky et al., 2009), the Long Range Arena (Tay et al., 2021) and the Speech Commands dataset (Warden, 2018). Our results (Tabs. 1, 2), demonstrate that CCNNs obtain state-of-the-art across several tasks, surpassing the performance of tailored architectures such as Recurrent Neural Networks and Transformers. The performance of CCNNs is explained by their ability to model long term dependencies with higher capabilities than tailored models. Interestingly, we observe that on tasks defined over flattened 2-dimensional signals, e.g., sMNIST, sCIFAR10,

Table 1: Experimental results on sequence, image and point-cloud datasets. × - *unable to apply, either due to the model's inability to handle the specified data, or to computational complexity. N.A. - not available.*

| | | SEQUENCE | | | | | | IMAGE | | | POINT-CLOUD |
|---|---|---|---|---|---|---|---|---|---|---|---|
| | SIZE | sMNIST | pMNIST | sCIFAR10 | SPEECH-MFCC | SPEECH-RAW | SPEECH-0.5x | CIFAR10 | CIFAR100 | STL10 | ModelNet40 |
| Transformer | 500K | 98.90 | 97.90 | 62.20 | 90.75 | × | × | × | × | × | × |
| LSSL | 7.8M | 99.53 | 98.76 | 84.65 | × | × | × | × | × | × | × |
| S4 | 7.8M | 99.63 | 98.70 | 91.13 | × | × | × | × | × | × | × |
| LSSL | 300k | N.A. | N.A. | N.A. | 93.58 | × | × | × | × | × | × |
| S4 | 300k | N.A. | N.A. | N.A. | 93.96 | 98.32 | 96.30 | × | × | × | × |
| CKCNN-Seq | 98K | 99.31 | 98.00 | 62.25 | 95.30 | 71.66 | 65.96 | × | × | × | × |
| CKCNN-Seq-Big | 1M | 99.32 | 98.54 | 63.74 | × | × | × | × | × | × | × |
| FlexTCN-6 | 375K | 99.62 | 98.63 | 80.82 | 97.67 | 91.73 | × | × | × | × | × |
| ResNet-44 | 660K | × | × | × | × | × | × | 92.90 | 71.15 | NA | × |
| ResNet-18 | 11.2M | × | × | × | × | × | × | 94.92 | 77.50 | 81.04 | × |
| ViT | 6.3M | × | × | × | × | × | × | 90.92 | 66.54 | N.A. | × |
| Swin-T/1 | 27.5M | × | × | × | × | × | × | 94.46 | 78.07 | N.A. | × |
| Parabolic CNN | 502k | × | × | × | × | × | × | 88.5 | 64.8 | 77.0 | × |
| Hamiltonian CNN | 264k | × | × | × | × | × | × | 89.3 | 64.9 | 78.3 | × |
| CKCNN | 630k | × | × | × | × | × | × | 86.8 | N.A. | N.A. | × |
| FlexNet-6 | 670k | × | × | × | × | × | × | 92.2 | N.A. | N.A. | × |
| SpiderCNN | N.A. | × | × | × | × | × | × | 77.97 | N.A. | N.A. | 92.4 |
| PointConv | N.A. | × | × | × | × | × | × | 89.3 | N.A. | N.A. | 92.5 |
| DGCNN | 1.84M | × | × | × | × | × | × | × | × | × | 89.0 |
| PointNet | 3.48M | × | × | × | × | × | × | × | × | × | 89.2 |
| PointNet++ | 1.99M | × | × | × | × | × | × | × | × | × | 90.0 |
| RepSurf-U | 1.48M | × | × | × | × | × | × | × | × | × | **94.7** |
| CCNN$_{4,140}$ | 200K | **99.72** | 98.82 | 90.30 | 95.01 | 98.34 | 96.22 | 92.78 | 66.86 | 81.80 | 84.44 |
| CCNN$_{6,380}$ | 2M | **99.72** | **98.84** | **93.08** | **97.98** | **98.44** | **96.44** | **95.20** | 73.16 | **83.00** | 85.70 |

CCNNs learn to construct very large periodic kernels that model flattened local dependencies in 2D (Fig. 12). This showcases the ability of CCNNs to model meaningful long range dependencies.

Additionally, we evaluate the capacity of CCNNs to classify speech (Warden, 2018) both from prepossessed (Speech-MFCC) and raw signals (Speech-Raw). Our results (Tab. 1) demonstrate state-of-the-art results on both preprocessed and raw signals –of length 16000– thus demonstrating the ability of CCNNs to process very heterogeneous data types with a unified architecture.

*Generalization across resolutions.* CCNNs are defined on a continuous space. Consequently, it is possible to train a CCNN at one resolution and deploy it at other resolutions –a feat not achievable with discrete models (Romero et al., 2022a; Nguyen et al., 2022)–. To showcase this ability, we train CCNNs on Speech-Raw, and test them on a subsampled version of the dataset (Speech-0.5x). CCNNs not only generalize, but outperform existing models on zero-shot prediction over resolution changes.

**CCNNs on image datasets.** In 2D, we consider the CIFAR10, CIFAR100 (Krizhevsky et al., 2009) and the STL10 datasets (Coates et al., 2011). CCNNs outperform existing continuous convolutional vision architectures (Ruthotto & Haber, 2020; Tomen et al., 2021; Romero et al., 2022b;a) and are competitive with existing large scale discrete models, while being (much) smaller (Tab. 1).

*The importance of modelling long range dependencies in $ND$.* In principle, all tasks can be treated as sequential tasks ignoring the $ND$ structure –as done in S4 (Gu et al., 2022) for images due to the complexity of defining state spaces on 2D–, but this sacrifices structural information. Contrarily, CCNNs can be easily defined on multidimensional spaces simply by changing the dimension of the input coordinates of the kernel generator networks. We observe that by considering the 2D structure of the Image and Pathfinder tasks of the LRA benchmark, much better results can be obtained (Tab. 2, right). In PathFinder with 2D images, the CCNN$_{6,380}$ obtains an accuracy of 96.00, outperforming the previous state-of-the-art by almost 10% points and performing remarkably better than on flattened images. Additionally, we observe that models trained on the original 2D data converge faster than their sequential counterparts (Fig. 9). Finally, we remark that discrete 2D CNNs with small convolutional kernels, e.g., ResNet-18 (He et al., 2016), are unable to solve Pathfinder due to the lack of fine-grained global context resulting from intermediate pooling layers. This was also seen by Gu et al. (2022).

**CCNNs on point-cloud datasets.** An additional advantage that comes from the continuous nature of CCNNs, is that –contrary to discrete CNNs– CCNNs can be seamlessly applied on irregular data, e.g., point-clouds. To assess the performance of CCNNs in 3D, we consider the ModelNet40 dataset (Wu et al., 2015), which consists of 3D meshes of objects often treated as point-clouds (Wu et al., 2019). Our results show that CCNNs are able to achieve a decent overall classification accuracy in comparison to point-cloud specific architectures with a relatively low parameter count (200 K). In particular, CCNNs outperform the Perceiver (Jaegle et al., 2021) while being much smaller (Fig. 8).

*From regular to irregular data.* Interestingly, the continuous nature of CCNNs allows us to use CCNNs trained on regular data to process irregular data. We showcase this ability by training a CCNN$_{4,140}$ on a version of ModelNet10 (Wu et al., 2015) voxelized onto a grid of 40×40×40 voxels –Fig. 11b shows an example of the point-cloud and voxelized datasets–. After trained, we deploy the CCNN on the original (point-cloud) and voxelized test sets and observe respective test accuracies of 90.99 and 90.64. This result shows that CCNNs learn representations that reflect the continuous

Table 2: Experimental results on the Long Range Arena benchmark. × - *unable to apply due to the model's inability to handle the specified data.*

|  | LISTOPS | TEXT | IMAGE | PATHFINDER | AVG. | 2DIMAGE | 2DPATHFINDER |
|---|---|---|---|---|---|---|---|
| Transformer | 36.37 | 64.27 | 42.44 | 71.40 | 53.66 | × | × |
| Reformer | 37.27 | 56.10 | 38.07 | 68.50 | 50.56 | × | × |
| BigBird | 36.05 | 64.02 | 40.83 | 74.87 | 57.17 | × | × |
| Linear Trans. | 16.13 | 65.90 | 42.34 | 75.30 | 50.46 | × | × |
| Performer | 18.01 | 65.40 | 42.77 | 77.05 | 51.18 | × | × |
| FNet | 35.33 | 65.11 | 38.67 | 77.80 | 54.52 | × | × |
| Nystromförmer | 37.15 | 65.52 | 41.58 | 70.94 | 57.46 | × | × |
| Luna-256 | 37.25 | 64.57 | 47.38 | 77.72 | 59.37 | × | × |
| S4 | **58.35** | 76.02 | 87.26 | 86.05 | **80.48** | × | × |
| $CCNN_{4,140}$ | 44.85 | 83.59 | 87.62 | 91.36 | 76.86 | 89.48 | 94.80 |
| $CCNN_{6,380}$ | 43.60 | **84.08** | **88.90** | **91.51** | 77.02 | **91.12** | **96.00** |

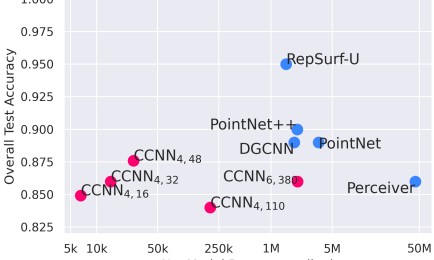

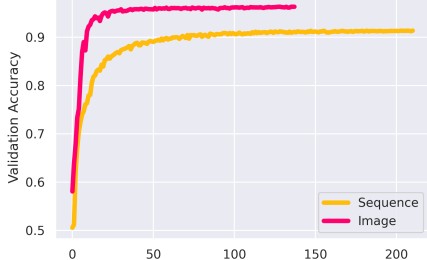

Figure 8: Parameter count versus performance on ModelNet40. Very small CCNN models obtain remarkable results.

Figure 9: Performance of the $CCNN_{6,380}$ on the LRA Image task, where samples are either input as sequence or with the original image structure.

nature of the data, showcasing CCNNs as a truly general-purpose architecture even able to generalize across different data representations –a feat hardly obtainable by existing architectures–.

*Large CCNNs overfit on point-clouds, but tiny CCNNs perform surprisingly well.* In contrast to 1D and 2D, we observe that larger CCNNs perform worse than smaller ones on 3D point-clouds (Fig. 8). We hypothesize that this is a result of the sparsity of the task. Continuous convolutional kernels define a kernel function over the entire domain even if the number of kernel values sampled is sparse. As a result, sparse supervision makes it hard for the network to learn a kernel function that generalizes to unseen data, as sparsity exposes the kernel function to aliasing –high frequencies between sampled points–. To corroborate this hypothesis, we train extremely small CCNNs –with 6K, 14K and 48K parameters– on ModelNet40. We observe that these models achieve surprising results and even outperform the 200K and 2M CCNNs (Fig. 8). These results supports our previous hypothesis, but also show that the continuous kernel paradigm can be used to construct extremely small, performant CNNs.

## 6 LIMITATIONS AND FUTURE WORK

**Computational efficiency.** Although convolutions with large convolutional kernels scale better than self-attention, e.g., Perceiver (Jaegle et al., 2021), they can still be expensive. Luckily, the Fourier convolution can be used to strongly reduce their cost, e.g., Romero et al. (2022b;a); Gu et al. (2022). Nevertheless, the Fourier convolutions by themselves are not sufficient to scale CCNNs to very large inputs. This problem is exacerbated if irregular data is considered, as it (*i*) prevents the usage Fourier convolutions, and (*ii*) requires rendering a different convolutional kernel for each spatial position of each sample in the batch. An important avenue for future research may look into reducing computational requirements, either via separability or self-adjusting architectures, e.g., downsampling (Riad et al., 2022), for regular data, and "gridifying" techniques for irregular data.

**Larger models on point-cloud data.** Although CCNNs performs remarkably well on point-clouds with extremely small models, the current CCNN formulation is unable to achieve better performance with larger models. We hypothesize that this is due to increasing aliasing in the learned kernel generator networks. Although anti-aliasing techniques for kernel generator networks exist (Romero et al., 2022a), this assume an underlying grid and a corresponding Nyquist frequency. An additional avenue for future research may look into the aliasing issue on irregular data either by generalizations of Romero et al. (2022a), or properly-defined smoothing techniques.

**Cross-modal training and data fusion.** CCNNs present a potential solution for cross-modal training –currently challenging due to dissimilar per-modality architectures–. Additionally, the continuous properties of CCNNs allow them to be trained on a wide range of data sources, e.g., multiple resolutions, regular, irregular data, etc, which is interesting for several machine learning application.

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

APPENDIX
# MODELLING LONG RANGE DEPENDENCIES IN $N$D: FROM TASK-SPECIFIC TO A GENERAL PURPOSE CNN

## A EXTENDED RELATED WORK

In this section, we provide a more extensive treatment of similar methods related to continuous reparameterizations of convolutional kernels and their limitations in use for general purpose architectures.

**Rethinking convolutional kernels.** Several previous works investigate a reformulation of CNNs based on modifications to the classical convolutional layer. These works provide formulations of continuous convolutional kernels pursuing several different motivations. Several works apply specifically to point-cloud data –for which discrete kernels are not appropriate– by interpolating a set of weights to obtain a definition over the continuous input space (Hua et al., 2018; Thomas et al., 2019) or expressing the kernels in a polynomial (Xu et al., 2018) or neural network basis (Jia et al., 2016; Schütt et al., 2017; Wang et al., 2018; Wu et al., 2019). These architectures often rely on sophisticated data augmentation and feature engineering schemes, perform pooling or downsampling operations, and have not been shown to perform well on regular data, e.g. images (Wu et al., 2019).

Other work focuses on continuous kernel formulations to address specific architectural considerations, for example to increase receptive fields efficiently (Su & Wen, 2021), or to learn more appropriate kernel geometries (Dai et al., 2017; Jeon & Kim, 2017). In these examples, a fixed set of weights is interpolated over the kernel domain, which implicitly defines a low-resolution kernel that impedes modeling long term dependencies in a dense manner (Romero et al., 2022a).

A different line of work incorporating continuous kernel definitions focuses on exploiting desirable properties of spatially structured kernels for implementing convolutions equivariant to roto-translations (Sifre & Mallat, 2014; Worrall et al., 2017; Weiler et al., 2018; Weiler & Cesa, 2019; Bekkers, 2019; Finzi et al., 2020), or dilation-translation transformations (Sosnovik et al., 2019; Worrall & Welling, 2019; Sosnovik et al., 2021). Although these formulations also in principle give us handles for a continuous definition of convolutional kernels over the spatial domain, we choose not to restrict the kernel functions learned in our models to be equivariant, as this limits applicability to settings in which such priors are not warranted.

Note that graph convolutions (Kipf & Welling, 2016) may similarly be viewed as providing a convolution operation invariant to arbitrary permutation symmetries on the input. Since this operation disregards positional information, it essentially shares the kernel function over the domain. In practice, the bias to permutation invariance needlessly impedes network expressivity for data where such assumptions do not hold, such as sequences, images or point-clouds.

Continuous Kernel Convolutions (Romero et al., 2022b) may be seen as a special case of neural message passing (Gilmer et al., 2017) on a fully connected graph over the input, with messages conditioned on relative positions of nodes. In the case of FlexConv (Romero et al., 2022a), graph connectivity is dynamic, conditioned on distance.

## B DATASET DESCRIPTION

**Sequential and Permuted MNIST.** The MNIST dataset LeCun et al. (1998) consists of 70K grayscale 28×28 handwritten digits divided into training validation and test sets of 60K and 10K samples, respectively. For validation purposes, the training dataset is further divided into training and validation sets of 55K and 5K samples, respectively.

The sequential MNIST dataset (sMNIST) presents MNIST images as a sequence of 784 pixels for digit classification. Consequently, good predictions require the model to preserve long-term dependencies up to 784 steps in the past. The permuted MNIST dataset (pMNIST) incorporates an additional level of difficulty by permuting the order of all sMNIST sequences with a random permutation. Resultantly, models can no longer rely on local information for the construction of their features and the importance of modelling long-term dependencies becomes more pronounced.

**CIFAR10, CIFAR100 and Sequential CIFAR10.** The CIFAR10 dataset Krizhevsky et al. (2009) consists of 60K real-world 32×32 RGB images uniformly drawn from 10 classes divided into training and test sets of 50K and 10K samples, respectively. The CIFAR100 dataset Krizhevsky et al. (2009) is similar to the CIFAR10 dataset, with the difference that the images are now uniformly drawn from 100 different classes. For validation purposes, the training dataset of both CIFAR10 and CIFAR100 are further divided into training and validation sets of 45K and 5K samples, respectively.

Analogously to the sMNIST dataset, the sequential CIFAR10 (sCIFAR10) dataset presents CIFAR10 images as a sequence of 1024 pixels for image classification. This dataset is more difficult than sMNIST, as *(i)* larger memory horizons are required to successfully solve the task, and *(ii)* more complex structures and intra-class variations are present in the images.

**Speech Commands.** The Speech Commands dataset Warden (2018) consists of 105809 one-second audio recordings of 35 spoken words sampled at 16kHz. Following Kidger et al. (2020), we extract 34975 recordings from ten spoken words to construct a balanced classification problem. We refer to this dataset as *Raw Speech Commands*. In addition, we use the preprocessing steps of Kidger et al. (2020) and extract mel-frequency cepstrum coefficients from the raw data. The resulting dataset, referred to as *MFCC Speech Commands*, consists of time series of length 161 and 20 channels.

**Long Range Arena.** The Long Range Arena benchmark (Tay et al., 2021) consists of 6 tasks with lengths 1K-16K steps encompassing modalities and objectives that require similarity, structural, and visuospatial reasoning. The `Pathfinder`, `Path-X` and `Image` tasks are similar in nature to the sMNIST and sCIFAR10 tasks. These tasks consists of classification tasks performed on images that are treated as sequences.

The `Image` task corresponds to the sequential CIFAR10 dataset with the only difference that the CIFAR10 images are treated as gray-scale images. The `Pathfinder` and `Path-X` tasks are binary tasks in which binary images are provided and the model must predict whether the two points in the images are connected with a line or not –see Fig. 10 for an example–. The difference between both datasets is their resolution. Whereas `Pathfinder` has images of size 32×32, `Path-X` has images of size 128×128. It is important to mention that these tasks are so difficult that even if treated as 2D signals, CNNs without global receptive fields cant solve them (Gu et al., 2022).

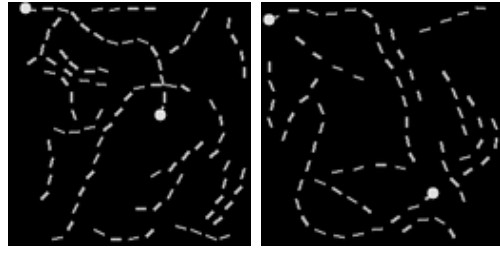

(a)        (b)

Figure 10: Positive and negative samples from the `Path-X` dataset

**STL-10.** The STL-10 dataset (Coates et al., 2011) is a subset of the ImageNet dataset (Krizhevsky et al., 2012) consisting of 13,000 96×96 real-world RGB images uniformly drawn from 10 classes divided into training and test sets of 5K and 8K images, respectively. For validation purposes, the training dataset is further divided into training and validation sets of 4,500 and 500 samples, respectively.

**ModelNet40.** The ModelNet40 dataset (Wu et al., 2015) contains 12,311 3D meshes of objects belonging to 40 classes. We use the official split with 9,843 training samples and 2,468 validation samples. In contrast with previous works which sample 1,024 points from these meshes, (e.g. Wu et al. (2019)), we sample 512 points uniformly from the faces of each mesh, along with the normal vectors at each of these positions. These position and normal vectors serve as input to the model.

**ModelNet10 Voxelized.** ModelNet10 is a subset of ModelNet40 containing orientation-aligned samples from 10 classes of objects. We voxelize ModelNet10 based on a subsampling of 4096 points from the meshes. First, we normalize these points to fall within the interval $[-1, 1]$. Next, we bin these points into a grid of $40 \times 40 \times 40$ voxels. All nonzero voxels get assigned their location as feature value. Afterwards, to align with our point-cloud configuration, we mask out the all but 512 nonzero voxels. Throughout the network, we

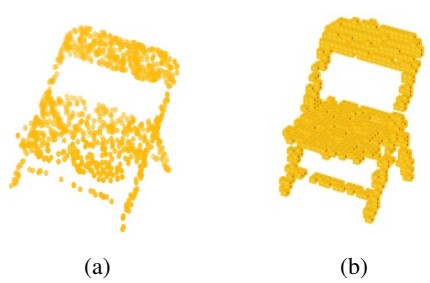

(a)        (b)

Figure 11: An example of a point-cloud sampled from ModelNet10 (a), and a corresponding voxel representation of the same sample (b).

only apply convolutions at the locations of each
nonzero voxel (effectively masking out activations
at all other voxel locations). Like with the point-
cloud configuration, to limit computational complexity, we limit the convolutions to integrate over
the closest 256 points.

## C  EXPERIMENTAL DETAILS

### C.1  GENERAL REMARKS

**Code repository and logging.** Our code is written in `PyTorch`. We utilize `wandb` (Biewald,
2020) `hydra` (Yadan, 2019) and `pytorch-lightning` (Falcon et al., 2019) for logging and
code structuring. Our experiments are performed on NVIDIA TITAN RTX, A6000 and A100 GPUs,
depending on the size of the datasets and inputs considered. Our code is publicly available at *url
hidden for the sake of the double blind review process.*

**Normalized relative positions.** The kernel network $\varphi_{\text{Kernel}}$ can, in principle, receive arbitrary
coordinates as input. However, considering unitary step-wise relative positions, i.e., 0, 1, 2, ... , N,
can be problematic from a numerical stability perspective as N may grow very large, e.g., N=16000
for the Speech Commands dataset. Consequently, based on insights from the Implicit Neural
Representations, e.g., Sitzmann et al. (2020); Fathony et al. (2021), we normalize the coordinates
such that they lie in the space $[-1, 1]^{\text{D}}$ for D-dimensional kernels. To this end, we map largest unitary
positions seen during training $[0, N]$ to a uniform linear space in $[-1, 1]$. Note that any possible
relative kernel positions outside of the trained kernel domain which may be encountered during
inference are masked out.

### C.2  HYPERPARAMETERS AND TRAINING DETAILS

**Optimizer and learning rate scheduler.** All our models are optimized with AdamW (Loshchilov &
Hutter, 2017) in combination with a cosine annealing learning rate scheduler Loshchilov & Hutter
(2016) and a linear learning rate warm-up stage of 10 epochs.

**Best hyperparameters found.** We perform hyperparameter search on the learning rate, dropout rate,
weight decay, and $\omega_0$ of our CCNNs for each task considered.[3] The best hyperparameters found are
reported in Tables 3 and 4.

**Parameter efficiency experiments on ModelNet40.** To further assess parameter efficiency of the
CCNN on ModelNet40, we run a number of experiments with smaller model sizes. Results for the
following models are shown in Fig. 8: $\text{CCNN}_{4,16}$ with 4 residual blocks, a channel size of 16 and a
kernel network hidden size of 8, with 6,545 total parameters. $\text{CCNN}_{4,32}$ with 4 residual blocks, a
channel size of 32 and a kernel network hidden size of 8, with 14,401 total parameters. $\text{CCNN}_{4,48}$
with 4 residual blocks, a channel size of 48 and a kernel network hidden size of 8, with 26,353 total
parameters. We use a learning rate of $2e^{-2}$, no weight decay, and an $\omega_0$ of 50.

## D  ADDITIONAL EXPERIMENTS AND DETAILS

### D.1  COMPUTATIONAL EFFICIENCY

### D.1.1  EFFICIENCY OF CONTINUOUS CONVOLUTIONAL KERNELS

**Experimental setup.** We experimentally asses the computational complexity of our model. We
measure the time of a single forward/backward pass for the two architectures used in our experiments:
$\text{CCNN}_{4,140}$ (4 blocks, 140 channels) and $\text{CCNN}_{6,380}$ (6 blocks, 380 channels). It is important to
note that we assume the "worst-case" scenario for CCNNs in which the learned kernel size equals
the input length, i.e., the model performs convolutions with global kernels at each layer. FlexConvs
(Romero et al., 2022a) learn the kernel size during training and –as shown exemplarily in Fig 12–
having global kernels at each layer is not something that we observe in practice.

---

[3] $\omega_0$ serves as a prior on the variance of the data that is fitted with several types of implicit neural representa-
tions, e.g., SIRENs Sitzmann et al. (2020), MFNs Fathony et al. (2021), etc.

Table 3: Best hyperparameters found for $\text{CCNN}_{4,140}$ on all tasks considered.

| | $w_0$ | Dropout | Learning Rate | Weight Decay | Batch Size | Epochs |
|---|---|---|---|---|---|---|
| SMNIST | 2976.49 | 0.1 | 0.01 | 1e-6 | 100 | 210 |
| PMNIST | 2985.63 | 0.2 | 0.02 | 0 | 100 | 210 |
| SCIFAR10 | 2386.49 | 0.0 | 0.02 | 0 | 50 | 210 |
| SPEECH COMMANDS (RAW) | 1295.61 | 0.2 | 0.02 | 1e-6 | 20 | 160 |
| SPEECH COMMANDS (MFCC) | 750.18 | 0.2 | 0.02 | 1e-6 | 100 | 110 |
| LISTOPS | 784.66 | 0.1 | 0.001 | 1e-6 | 50 | 60 |
| TEXT | 2966.60 | 0.2 | 0.001 | 1e-5 | 50 | 60 |
| IMAGE | 4005.15 | 0.2 | 0.01 | 0 | 50 | 210 |
| PATHFINDER | 2272.56 | 0.2 | 0.01 | 0 | 100 | 210 |
| CIFAR10 | 1435.77 | 0.1 | 0.02 | 0.0001 | 50 | 210 |
| CIFAR100 | 3521.55 | 0.1 | 0.02 | 0.0001 | 50 | 210 |
| STL10 | 954.28 | 0.1 | 0.02 | 0 | 64 | 210 |
| 2DIMAGE | 2085.43 | 0.2 | 0.02 | 1e-6 | 50 | 210 |
| 2DPATHFINDER | 1239.14 | 0.1 | 0.01 | 0 | 100 | 210 |
| MODELNET40 | 50 | 0.0 | 0.02 | 1e-8 | 16 | 200 |

Table 4: Best hyperparameters found for $\text{CCNN}_{6,380}$ on all tasks considered.

| | $w_0$ | Dropout | Learning Rate | Weight Decay | Batch Size | Epochs |
|---|---|---|---|---|---|---|
| SMNIST | 2976.49 | 0.1 | 0.01 | 0 | 100 | 210 |
| PMNIST | 2985.63 | 0.2 | 0.02 | 0 | 100 | 210 |
| SCIFAR10 | 4005.15 | 0.25 | 0.01 | 0 | 50 | 210 |
| SPEECH COMMANDS (RAW) | 1295.61 | 0.2 | 0.02 | 1e-6 | 20 | 160 |
| SPEECH COMMANDS (MFCC) | 750.18 | 0.2 | 0.02 | 1e-6 | 100 | 110 |
| LISTOPS | 784.66 | 0.25 | 0.001 | 0 | 50 | 60 |
| TEXT | 2966.60 | 0.3 | 0.02 | 0 | 50 | 60 |
| IMAGE | 4005.15 | 0.1 | 0.01 | 0 | 50 | 210 |
| PATHFINDER | 2272.56 | 0.1 | 0.01 | 1e-6 | 100 | 210 |
| CIFAR10 | 1435.77 | 0.15 | 0.02 | 0 | 50 | 210 |
| CIFAR100 | 679.14 | 0.2 | 0.02 | 0 | 50 | 210 |
| STL10 | 954.28 | 0.1 | 0.01 | 1e-6 | 64 | 210 |
| 2DIMAGE | 2306.08 | 0.2 | 0.02 | 0 | 50 | 210 |
| 2DPATHFINDER | 3908.32 | 0.2 | 0.01 | 0 | 100 | 210 |
| MODELNET40 | 50 | 0.0 | 0.02 | 1e-7 | 32 | 200 |

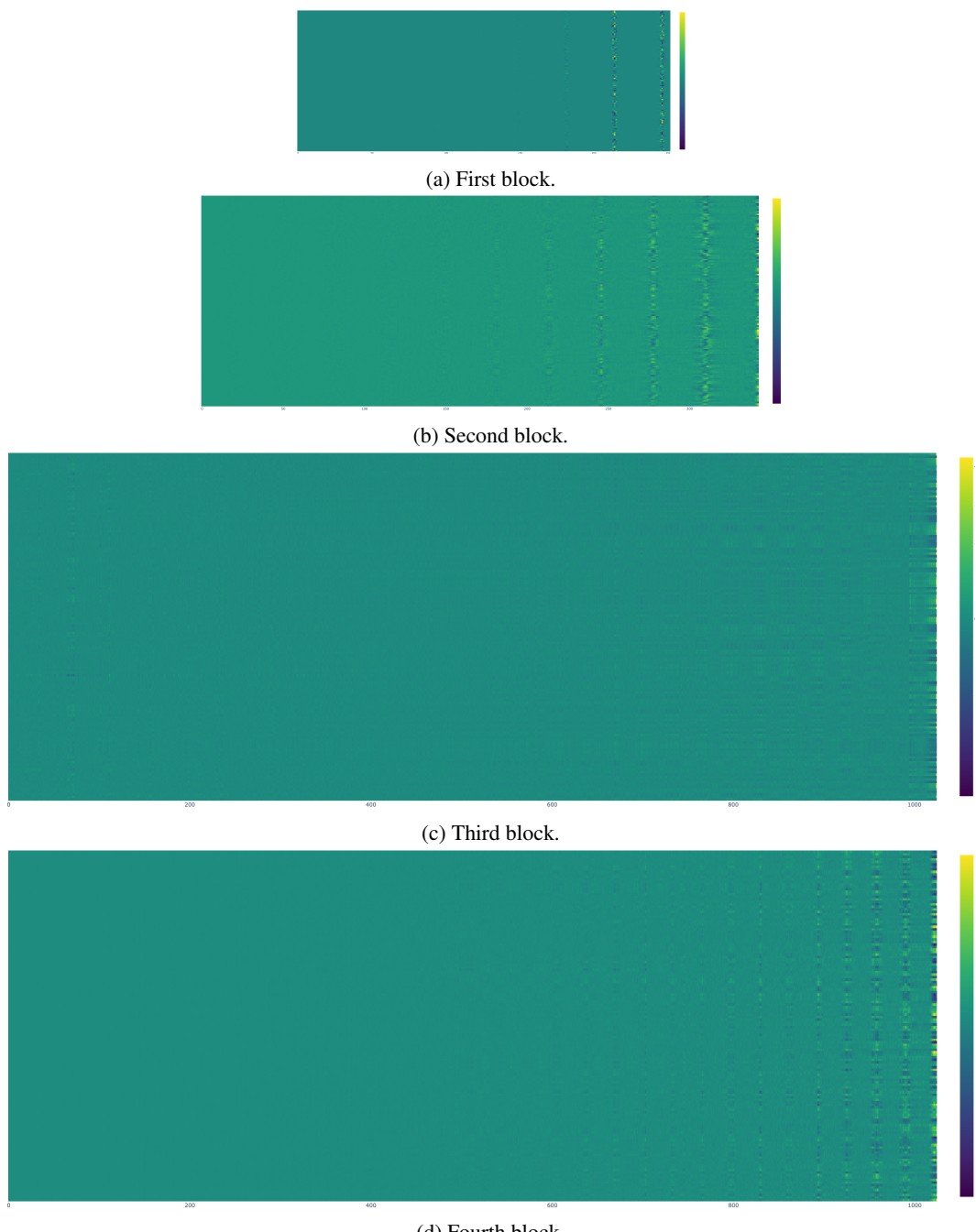

(a) First block.

(b) Second block.

(c) Third block.

(d) Fourth block.

Figure 12: Learned (causal) convolutional kernels in a CCNN$_{4,140}$ trained on sequential CIFAR10. The vertical axis indexes the channels, and the horizontal axis the length of the kernels in a single layer. Interstingly, we observe a clear periodic pattern of 32 steps along the spatial dimension across all layers –a period that corresponds to the width of the underlying 32×32 CIFAR10 images–. This illustrates that CCNNs in fact learn to represent 2D structures on the flattened 1D space on which sequential CIFAR10 is defined. Despite these important capabilities, it is important to note that modelling $N$D signals as flattened 1D signals poses a unnecessary burden to the model, and modelling the signal in the original 2D space leads to faster convergence and better accuracy (Fig. 9).

Table 5: Training time (ms). × - *unavailable due to memory issues.* $n$ - *number of blocks tuned per input length.*

| SIZE | 1024 | 4096 | 8192 | 16000 |
|---|---|---|---|---|
| Local CNN$_{n,140}$ | 24 | 32 | 43 | 65 |
| Global CNN$_{4,140}$ | 13 | 25 | 53 | 122 |
| CCNN$_{4,140}$ | 52 | 56 | 77 | 145 |
| Local CNN$_{n,380}$ | 24 | 38 | 93 | 178 |
| Global CNN$_{6,380}$ | 31 | 102 | 220 | × |
| CCNN$_{6,380}$ | 71 | 136 | 255 | 537 |

We compare to (*i*) discrete convolutional networks which match in architecture to ours, with global depthwise separable convolutional kernels; Global CNN$_{4,140}$, and Global CNN$_{6,380}$, (*ii*) more traditional CNN architectures with downsampling after each block (pooling of size 2), local depthwise separable kernels (kernels of size 5), and a depth tuned to the input size such that the receptive field of the final residual block covers the entire input. We take $\text{depth} = \lceil d \rceil$ with d given by:

$$\frac{\text{input length}}{\text{pooling size}^{\text{d}}} \geq \text{kernel size}.$$

These architectures are denoted Local CNN$_{4,140}$ and Local CNN$_{6,380}$. For input lengths of 1024, 4096, 8192 and 16000 this results in depths of 8, 10, 11 and 12 blocks respectively.

We measure training time (forward and backward pass) for sequence data of different lengths between 1024 –the length of sCIFAR10–, and 16000 –the length of Speech-Raw–, the largest sequence length in our experiments. We measure the time of training over 1000 batches of size 8 on a RTX 3090.

**Results**. Results are summarized in Tab 5. Note that for the Global CNN$_{6,380}$, we were unable to complete the experiment on data of lengths exceeding 8192 because of insufficient memory during the backward pass. This highlights another benefit of the continuous convolutional layer: its relative memory efficiency in computing weight updates, which results from the fact that gradients for individual kernel values can be discarded. As only the kernel network weights need to be updated, gradient values for the kernel weights do not need to be stored during the backward pass.

Our results show that our approach requires additional computation compared to more traditional discrete CNNs. We note that although the use of a kernel network to infer kernel values brings computational overhead, this overhead is not a limiting factor in practice. The global convolutional kernels used in our network architectures have an impact on computational efficiency, but the use of convolutions in the Fourier domain help address this issue.

### D.1.2 EFFICIENCY OF CCNNS WITH REGARD TO OTHER GLOBAL METHODS

**Experimental Setup.** Next, we benchmark the computational efficiency of CCNNs compared to other input-global methods: S4 (Gu et al., 2022) and LSSL (Gu et al., 2021).

We reproduce the efficiency benchmark in Gu et al. (2022). That is, we provide the runtime for a single layer of forward and backward pass for different hidden dimensionalities, measured for different input lengths. The kernel network used in these experiments is a MAGNet with 3 layers and a hidden dimensionality of 32. Again, we assume a "worst-case" setting in which the learned kernel size equals the input length, i.e. the layer performs convolutions with global kernels. Note that in this setting, we perform convolutions in the spatial domain. We average runtime over 10000 steps. Experiments are performed on a RTX 3090.

**Results**. Results are summarized in Tab. 6. We show that, up to a sequence length of 16000, runtimes in comparison with S4 are marginally faster, but remain in the same order of magnitude. Differences in memory utilization become more pronounced for increases in hidden dimensionality and sequence length, but remain a constant factor. Next, note that in both memory and speed, we show that a CCNN layer is asymptotically more efficient than LSSL. For LSSL, only results present in Gu et al. (2022) are shown, as we were unable to successfully reproduce the LSSL layer performance.

Table 6: Forward and backward for a single layer of CCNN in comparison with S4 (Gu et al., 2022) and LSSL (Gu et al., 2021) in speed (ms). × - *result not available.*

| INPUT LENGTH | 128 | | | 1024 | | | 4096 | | | 16000 | | |
|---|---|---|---|---|---|---|---|---|---|---|---|---|
| HIDDEN SIZE | 256 | 512 | 1024 | 256 | 512 | 1024 | 256 | 512 | 1024 | 256 | 512 | 1024 |
| LSSL | 9.32 | 20.60 | 140.70 | × | × | × | × | × | × | × | × | × |
| S4 | 4.16 | **3.41** | 4.16 | 4.11 | 6.39 | 9.06 | 6.86 | 10.57 | 25.20 | **18.10** | **35.72** | **96.64** |
| CCNN | **3.99** | 3.71 | **4.12** | **3.90** | **5.07** | **7.25** | **5.91** | **9.67** | **22.39** | 20.64 | 42.49 | 102.57 |

Table 7: Forward and backward for a single layer of CCNN in comparison with S4 (Gu et al., 2022) and LSSL (Gu et al., 2021) memory usage (Mb). × - *result not available.*

| INPUT LENGTH | 128 | | | 1024 | | | 4096 | | | 16000 | | |
|---|---|---|---|---|---|---|---|---|---|---|---|---|
| HIDDEN SIZE | 256 | 512 | 1024 | 256 | 512 | 1024 | 256 | 512 | 1024 | 256 | 512 | 1024 |
| LSSL | 222.1 | 1685 | 13140 | × | × | × | × | × | × | × | × | × |
| S4 | **5.6** | **14.46** | **42.64** | **33.65** | **70.47** | **153.69** | **129.69** | **262.51** | **537.72** | **508.93** | **1014.08** | **2033.82** |
| CCNN | 14.46 | 28.94 | 60.38 | 113.13 | 222.97 | 444.17 | 451.39 | 888.24 | 1763.43 | 1764.46 | 3467.34 | 6878.26 |

## D.2  ABLATION: PERFORMANCE OF FLEXCNN, S4 AND CCNN RESIDUAL BLOCKS

**Experimental setup**. We provide an ablation over the impact on performance of the proposed residual CCNNBlock compared to the FlexCNNBlock used in the architectures in Romero et al. (2022a) and the S4Block used in Gu et al. (2022) (Fig. 7).

To this end, we run experiments on a select number of datasets covering sequence and image data with a version of our architecture that includes the FlexCNNBlock (F-CCNN$_{4,140}$ with 4 blocks, 140 channels, 233K params and F-CCNN$_{6,380}$ with 6 blocks, 380 channels, 2.24M params), and a version of our architecture that includes the S4Block (S4-CCNN$_{4,140}$ with 4 blocks, 140 channels, 200K params and S4-CCNN$_{6,380}$ with 6 blocks, 380 channels, 2M params). We compare performance against the two architectures used throughout the experiments in Sec. 5, which uses the CCNNBlock (CCNN$_{4,140}$ with 4 blocks, 140 channels, 200K params and CCNN$_{6,380}$ with 6 blocks, 380 channels, 2M params). To isolate the impact of the residual block architecture, we replace the FlexConv layers in the original formulation of the FlexCNNBlock with our proposed depthwise separable implementation SepFlexConv. Note that we do not parameter-match these models, which results in the architectures with FlexCNNBlocks having more parameters compared to ones with the CCNNBlock, due to the FlexCNNBlock having two convolutional layers instead of the one convolutional layer and one pointwise linear layer of the CCNNBlock. Training regimes and hyperparameters for all architectures are as per Sec. C.

**Results**. Results are summarized in table 8. First, note that without the nonlinearity added add the end of the block, the S4Block performs poorly over all tasks. Next, as shown, the CCNNBlock formulation improves performance of the architecture compared to FlexCNNBlock, in some cases by a significant margin. Note that architectures with the S4Block and CCNNBlock contain fewer convolutional layers and parameters compared to architecturally matched ones with FlexCNNBlocks, indicating higher parameter-efficiency as well as computational efficiency of CCNNBlocks (e.g. 8h 55m runtime for F-CCNN$_{6,380}$ vs. 8h 12m runtime for CCNN$_{6,380}$ on sCIFAR10). Because of the shown performance benefits, we use the CCNNBlock throughout our experiments.

Table 8: Test accuracy of models with FlexCNNBlock (F-CCNN$_{4,140}$, F-CCNN$_{6,380}$), models with S4Block (S4-CCNN$_{4,140}$, S4-CCNN$_{6,380}$) and models with CCNNBlock (CCNN$_{4,140}$, CCNN$_{6,380}$).

| | SPEECHCOMMANDS-MFCC | sCIFAR10 | CIFAR10 |
|---|---|---|---|
| F-CCNN$_{4,140}$ | 94.38 | 85.54 | 86.34 |
| S4-CCNN$_{4,140}$ | 73.60 | 53.25 | 60.06 |
| CCNN$_{4,140}$ | **95.01** | **90.30** | **92.78** |
| F-CCNN$_{6,380}$ | 94.92 | 84.68 | 92.24 |
| S4-CCNN$_{6,380}$ | 60.82 | 51.50 | 67.01 |
| CCNN$_{6,380}$ | **97.98** | **93.08** | **95.20** |

