# OpenReview forum: "Modelling Long Range Dependencies in $N$D: From Task-Specific to a General Purpose CNN"
_ICLR.cc/2023/Conference — ICLR 2023 poster_

### Official Review · Reviewer_RGNG · 2022-10-17

**Confidence:** 4
**Correctness:** 3
**Technical Novelty And Significance:** 2
**Empirical Novelty And Significance:** 3
**Recommendation:** 6

**Clarity, Quality, Novelty And Reproducibility:**

I personally think the clarity and reproducibility of this paper are ok. However, the current manuscript has some overlaps with the existing works [2], which makes the real contribution of this paper very limited. A thorough discussion of the differences between CCNN and CKConv would help to make a fair judgment of this submission.

# Reference:
[2] CKConv: Continuous kernel convolution for sequential data. ICLR2022

**Strength And Weaknesses:**

# Strength:
- The idea of "a CNN architecture that can be used across input resolutions, lengths and dimensionalities (1D, 2D, 3D) showing its viability across several 1D, 2D and 3D tasks" seems interesting, which may facilitate the cross-modal pre-training to further enhance the performance of large neural networks.
- The experimental results seem encouraging that CCNN tends to outperform strong baselines on sequence data, e.g., the competitive results on the Long Range Arena dataset.

# Weaknesses:
- ### Major concerns:
  - I politely disagree that "Convolutional Neural Network (CNN) architectures must be tailored to specific tasks in order to consider the length, resolution, and dimensionality of the input data" since the input data size can be inconsistent with the training dataset. Though CNN can not be directly applied to the 1D/3D input when pre-trained on 2D images, the descriptions of CNN as "current CNN architectures are resolution-bounded and thus different resolutions require different CNNs" and "there is no trivial way to obtain equivalent kernel values on upsampled data, and hence, no way to apply a trained model directly" seem inappropriate.
  - The authors claim that "our Continuous Convolutional Neural Network parameterizes kernel values as a continuous function $\varphi$ over the input domain $R^d$, which decouples it from data characteristics". It would be better to further discuss the "data characteristics" decoupled by the continuous function. Besides, it is true that a continuous kernel is compatible with arbitrary size/resolution/dimensionality, however, given the limited computing resources, a sampling operation is required during pre-processing (also pointed out by the authors in **Computational efficiency**). Is "informed sampling" against the motivation of a data-independent network?
  - Though the authors elaborate on the "data-independent" architecture, I am not fully convinced that this setting should contribute to higher performance. Notice that the popular Transformer architecture still requires conventional convolution operations to extract features and then conducts multi-head self-attention. From my point of view, convolution operations that introduce inductive bias could be more suitable for images. Then, the authors may further discuss the motivation of this paper.
  - It would be better to report the real runtime speed instead of the model parameters to make a fair comparison with S4 [1].

- ### Minor Comments:
  - **Parameter scaling**: the description of "CCNN formulation lacks in parameter scaling capabilities; increased parameter counts yield strongly
diminishing returns" seems unclear.
  - **Separable Continuous Convolutions**: How to determine the input channel number and the feature map size of the continuous "spatial" separable convolutional layer? It would be better to explain the meaning of $D$, $N_{in}$, and $N_{out}$.
  - **Table 1**: As claimed in *Empirical results*, "the CCNN is not restricted to grid data, e.g., 3D voxels, and can be used on point-clouds directly", I expect promising results on ModelNet40. However, the $CCNN_{4,110}$ seems far from satisfactory.
  - I am curious why the authors did not increase the weight decay $\lambda$ given that "we observed strong overfitting to the training set, which
may indicate the need for stronger regularization"?
  - **Normalized relative positions**: What if the largest unitary position is larger than $N$ at inference time?
  - **Code repository and logging**: It seems that the code link is missing.
  - **Section 3.1**: "Pointwise operations are applied independently to each spatial element of the input signal". However, each spatial element shares the same parameter for 1x1 convolutions.
  - Please fix the "Sec.??" on page 4.
  - I expect an ablation study on the effect of the S4 Block [1].

# Reference:
[1] Efficiently modeling long sequences with structured state spaces. ICLR2022


**Summary Of The Paper:**

The authors present a Convolutional Neural Network (CNN) architecture based on pointwise operations, global operations, and local operations, i.e., continuous convolutional kernels to handle the input with different lengths and resolutions. They show that this setting is data independent, which models long-range dependencies at every layer, with a fixed number of parameters. The experimental results show that the proposed Continuous Convolutional Neural Network (CCNN) performs well on the Long Range Arena dataset given sequence data.

**Summary Of The Review:**

Overall, my major concern is the real novelty of this paper. Besides, considering that the empirical results on several datasets are relatively lower than the baseline results, the effectiveness of the proposed method remains unclear. The authors are encouraged to solve the weaknesses above.

---

> ### Author Response · Authors · 2022-11-14
> **First response -- reviewer RGNG**
>
> Dear reviewer RGNG,
>
> First of all, we would like to thank you very much for your thorough review. We sincerely appreciate the time you spent in evaluating our work, and very much appreciate your comments.
>
> Here we will answer your questions, comments and concerns:
>
> **I politely disagree that "Convolutional Neural Network (CNN) architectures must be tailored to specific tasks in order to consider the length, resolution, and dimensionality of the input data" since the input data size can be inconsistent with the training dataset.**
>
> It is true that datasets can have inputs of different length, e.g., Text task in LRA. However, this is easily solved similarly to the way it is solved with conventional CNNs and other models. For a given dataset, we can compute the maximum length and make that the maximum length (which in our case is mapped to [-1, 1] ). Then, if smaller inputs are then provided, then it simply is mapped to the corresponding length (in fact for batching purposes, inputs in a batch are normally zero padded to the same length in order to treat them as a batch of the same length). The same procedure can be done with CCNNs.
>
> What we refer to with that *“architectures must be tailored to specific tasks in order to consider the length, resolution, and dimensionality of the input data"* is that if a CNN is created to model images (or sequences) of a certain length, then the same network cannot be generally used for other input lengths without structural changes. This is because pooling layers as well as depths must be tuned for them to be able to model long term dependencies with small (discrete) convolutional kernels. CCNNs on the other hand, are able to model long term dependencies at every layer, and thus do not require different tuning for inputs of different length, resolution or dimensionality. Some examples are provided in Sec. 3.
>
> **Though CNN can not be directly applied to the 1D/3D input when pre-trained on 2D images, the descriptions of CNN as "current CNN architectures are resolution-bounded and thus different resolutions require different CNNs" and "there is no trivial way to obtain equivalent kernel values on upsampled data, and hence, no way to apply a trained model directly" seem inappropriate.**
>
> Please let us know if the previous answer clarifies this concern.
>
> **The authors claim that "our Continuous Convolutional Neural Network parameterizes kernel values as a continuous function φ over the input domain Rd, which decouples it from data characteristics". It would be better to further discuss the "data characteristics" decoupled by the continuous function.**
>
> We appreciate the reviewer’s concerns regarding clarity on the term “data characteristics”. We now define these properly. For completeness, with data characteristics we refer to the three characteristics conventional CNNs are bound to: the length, resolution and dimensionality of the input.

---

> > ### Author Response · Authors · 2022-11-14
> > **First response -- reviewer RGNG -- continuation**
> >
> > **Besides, it is true that a continuous kernel is compatible with arbitrary size/resolution/dimensionality, however, given the limited computing resources, a sampling operation is required during pre-processing (also pointed out by the authors in Computational efficiency). Is "informed sampling" against the motivation of a data-independent network?**
> >
> > We believe that this question is a result of some misunderstandings regarding our limitations section. We unfortunately did not include the last version of those in our submission and the limitations outlined in our submission were only relating to the point-cloud experiments. This has been modified now in our updated manuscript. We sincerely apologise for this mistake.
> >
> > This sampling procedure is a common preprocessing step taken from previous works [1-4] and used to describe meshes as point-clouds. Do note that this is not necessary for datasets with regular data (in our work all 1D and 2D datastes). Objects in the ModelNet40 dataset are defined as meshes and point-clouds are obtained from these meshes via informed sampling. Nevertheless, it is true that for 3D (this is not the case in 1D and 2D due to batching) we must limit the number of points used during the convolution operation. This is needed because for each point in the point-cloud, a distance to each other point in the point-cloud must be computed in order to render the kernel. Nevertheless,  this does not tie the network architecture to data resolution, length or dimensionality; essentially the subsampling could be seen as a dropout operation on the feature maps, leading to an approximation of the convolution integral with smaller subsampling sets leading to increased variance within network activations [3].
> >
> > Indeed, this points to a limitation of the proposed work, which we discuss in our Limitations section and for which we propose some possible future research directions.
> >
> > [1] Qi, C. R., Su, H., Mo, K., & Guibas, L. J. (2017). Pointnet: Deep learning on point sets for 3d classification and segmentation. In Proceedings of the IEEE conference on computer vision and pattern recognition (pp. 652-660).
> >
> > [2] Qi, C. R., Yi, L., Su, H., & Guibas, L. J. (2017). Pointnet++: Deep hierarchical feature learning on point sets in a metric space. Advances in neural information processing systems, 30.
> >
> > [3] Wu, W., Qi, Z., & Fuxin, L. (2019). Pointconv: Deep convolutional networks on 3d point clouds. In Proceedings of the IEEE/CVF Conference on Computer Vision and Pattern Recognition (pp. 9621-9630).
> >
> > [4]Xu, Y., Fan, T., Xu, M., Zeng, L., & Qiao, Y. (2018). Spidercnn: Deep learning on point sets with parameterized convolutional filters. In Proceedings of the European Conference on Computer Vision (ECCV) (pp. 87-102).

---

> > > ### Author Response · Authors · 2022-11-14
> > > **First response --reviewer RGNG -- continuation 2**
> > >
> > > **Though the authors elaborate on the "data-independent" architecture, I am not fully convinced that this setting should contribute to higher performance. Notice that the popular Transformer architecture still requires conventional convolution operations to extract features and then conducts multi-head self-attention. From my point of view, convolution operations that introduce inductive bias could be more suitable for images. Then, the authors may further discuss the motivation of this paper.**
> > >
> > > We would like to emphasise that we do not oppose employing inductive biases in network design. On the contrary, this is why we use convolutions and not other types of universal architectures, e.g.Transformers, MLPs. Using convolutions make our networks translation equivariant –as other CNNs–, and thus respect their inductive biases. In addition, we would like to point out that our model actually includes an additional inductive bias compared to conventional CNNs by requiring the convolutional kernel to be continuous, which is a desirable property as it reflects the continuous nature of Euclidean data (audio, images, point-clouds).
> > >
> > > The contribution of our paper is that it decouples architecture design from parameters that do not reflect any task-specific inductive bias (kernel size, pooling, depth), but that rather are necessary to account for the fact that conventional CNNs do not respect the continuous nature of the data, and thus are unable to model long term dependencies at every layer. Note that many differences between task-specific CNN architectures are not a result of the inclusion of inductive biases into model design, but rather an artefact of the discrete nature of convolutional kernels used in them. For example, the need for more pooling operations in architectures applied to higher resolution images does not reflect any specific inductive bias, but it is simply included to obtain an adequate increase in receptive field in order to model dependencies over the increased spatial resolution of the input with discrete convolutional kernels.
> > >
> > > We propose to move to a paradigm which removes the need for these arbitrary choices in architectural hyperparameters; a higher resolution input image should not be reflected by architectural modifications, as the network should model the same underlying function only in higher fidelity.
> > >
> > > In summary, the proposed unified framework improves transferability of insights in network architecture design and reduces the need for task-specific network modifications that do not reflect valuable inductive biases. The value of our work lies in proposing a unification for conventional CNNs which are normally separated by seemingly arbitrary hyperparameter choices (e.g. kernel size / resolution, network depth, number of pooling operations). This viewpoint leads to a simplified architectural design without performance loss, which we validate by obtaining state-of-the-art performance on multiple datasets.
> > >
> > > **It would be better to report the real runtime speed instead of the model parameters to make a fair comparison with S4.**
> > >
> > > We agree that inference speed is an important consideration in model comparison. To add transparency to our manuscript, we added experiments on the computation complexity of our method (Appx. D.1). We show computational overhead in training speed compared to discrete CNNs, but that remains in similar order of magnitude. Moreover, we compare the layers in our network against the efficient S4 layer [1] and show similar performance in terms of speed (although CCNN is marginally faster for input lengths up to 8192). Our use of convolutional layers through Fourier convolutions does lead to increased memory usage, but differences remain a constant factor. Compared to LSSL [2], we show to be asymptotically more efficient both in terms of speed and memory usage.
> > >
> > > [1] Efficiently modeling long sequences with structured state spaces. ICLR2022
> > >
> > > [2] Gu, A., Johnson, I., Goel, K., Saab, K., Dao, T., Rudra, A., & Ré, C. (2021). Combining recurrent, convolutional, and continuous-time models with linear state space layers. Advances in neural information processing systems, 34, 572-585.
> > >
> > > **Parameter scaling: the description of "CCNN formulation lacks in parameter scaling capabilities; increased parameter counts yield strongly diminishing returns" seems unclear.**
> > >
> > > This is again due to our mistake with regard to the limitations section. We sincerely apologise for that.
> > >
> > > We meant to say that, for point cloud models, additional parameters did not seem to improve accuracy very much. In fact we observed a decay in performance for larger models (87.3% acc with 48k params, 85.7% acc with 2M params). However, we would like to note, that CCNNs still outperform the Perceiver –the most well known general purpose architecture– on point-clouds.
> > >  We have now described why this problem appears in our limitation section, and we intend to address these limitations in future work.

---

> > > > ### Author Response · Authors · 2022-11-14
> > > > **First response -- reviewer RGNG -- continuation 3**
> > > >
> > > > **Separable Continuous Convolutions: How to determine the input channel number and the feature map size of the continuous "spatial" separable convolutional layer? It would be better to explain the meaning of D, Nin, and Nout.**
> > > >
> > > > D refers to data dimensionality, Nin refers to the number of input channels and Nout refers to the number of output channels. We have clarified this in the manuscript.
> > > >
> > > > **Table 1: As claimed in Empirical results, "the CCNN is not restricted to grid data, e.g., 3D voxels, and can be used on point-clouds directly", I expect promising results on ModelNet40. However, the CCNN4,110 seems far from satisfactory.**
> > > >
> > > > It is true that the results of the CCNN_{4,110} and CCNN_{6, 380} are relatively poor compared to state-of-the-art point-cloud processing methods. However, we do point out that CCNNs are still better than the Perceiver --the best general purpose architecture to date-- when applied on point-clouds [1].
> > > >
> > > > [1] Jaegle A, Gimeno F, Brock A, Vinyals O, Zisserman A, Carreira J. Perceiver: General perception with iterative attention. InInternational conference on machine learning 2021 Jul 1 (pp. 4651-4664). PMLR.
> > > >
> > > > **I am curious why the authors did not increase the weight decay λ given that "we observed strong overfitting to the training set, which may indicate the need for stronger regularization"?**
> > > >
> > > > This is again due to our mistake with regard to the limitations section. We sincerely apologise for that.
> > > >
> > > > Using weight decay on 1D and 2D did have the intended results. However, on point-clouds an increased weight decay did not improve performance of CCNN_{4,110} and CCNN_{6,380} on point-clouds. We do observe improved performance on point-clouds with much smaller models in section *“CCNN for parameter efficient point-cloud processing”*, where we report 87.5% accuracy with 48k trainable parameters. We regard this as noteworthy compared to point-cloud specific methods such as pointnet and pointnet++ which obtain 89.0% and 89.2% accuracy with 3.48M and 2M trainable parameters respectively, considering our method requires no architectural changes going from sequence to image to point-cloud data.
> > > >
> > > > **Normalized relative positions: What if the largest unitary position is larger than N at inference time?**
> > > >
> > > > This is not a problem. If the input length is larger than the maximum sequence length observed during training, we can simply pass coordinates larger than 1 to generate kernels for that particular position. Note however that if the length of the input changes due to resolution changes, it is better to sample the normalized coordinates in a finer grid. This grants CCNNs the ability to generalize to unseen resolutions.
> > > >
> > > > Nevertheless, a valid question is whether the model is able to extrapolate to unseen lengths. Honestly, we did not consider that setting in our paper, but we are aware that this is an open research problem even for large language models. Investigating whether the continuous kernel parameterization can be used for extrapolating models, e.g., for language models, reinforcement learning,  is an interesting direction for future research.
> > > >
> > > > **Code repository and logging: It seems that the code link is missing.**
> > > >
> > > > The code has been anonymized and is now submitted as supplementary material. Moreover, we would like to note that our code is already publicly available, but we have removed the link from the submission in order to respect the double-blind review process.
> > > >
> > > > **Section 3.1: "Pointwise operations are applied independently to each spatial element of the input signal". However, each spatial element shares the same parameter for 1x1 convolutions.**
> > > >
> > > > We understand the concern regarding the phrasing here, indeed use of the word “independently” can be ambiguous. What we meant to say here is that each position is processed independently, i.e., it does not depend on its neighbours. We have amended this in our manuscript.
> > > >
> > > > **Please fix the "Sec.??" on page 4.**
> > > >
> > > > This has been fixed.
> > > >
> > > > **I expect an ablation study on the effect of the S4 Block [1].**
> > > >
> > > > Based on the reviewer’s suggestion, we now provide an ablation over the residual block used in our architecture, and compare it to the residual blocks used in [1] and [2]. In addition, we have improved Section 4 in terms of readability and illustrations (the different types of blocks as well as the final CCNN architecture are now shown in the main text).  With regard to the ablation, the experimental setup and results are provided in Appx D.2. We show significant increases in performance owing to the architectural changes in the residual block. This illustrates why the proposed changes were made.
> > > >
> > > > [1] Efficiently modeling long sequences with structured state spaces. ICLR2022
> > > >
> > > > [2] Romero, D. W., Bruintjes, R. J., Tomczak, J. M., Bekkers, E. J., Hoogendoorn, M., & van Gemert, J. C. (2021). Flexconv: Continuous kernel convolutions with differentiable kernel sizes. arXiv preprint arXiv:2110.08059.

---

> > > > > ### Author Response · Authors · 2022-11-14
> > > > > **First response -- reviewer RGNG -- continuation 4**
> > > > >
> > > > > **I personally think the clarity and reproducibility of this paper are ok. However, the current manuscript has some overlaps with the existing works [2], which makes the real contribution of this paper very limited. A thorough discussion of the differences between CCNN and CKConv would help to make a fair judgment of this submission.**
> > > > >
> > > > > We have strongly improved the clarity and reproducibility of our submission. We have made several adjustments to illustrate better the contributions of our work and what sets it apart from existing ones. In particular, we have (i) changed the title, (ii) added a contributions subsection to the introduction and (iii) restructured Sec. 4. We hope that the current form of our paper highlights our contributions more clearly and properly sets our work apart from previous work.
> > > > >
> > > > > With regard to reproducibility, we have included an anonymized version of the code. In addition, we note that the code is already publicly available, but the link has not been included to respect the double blind review process.
> > > > >
> > > > > **Besides, considering that the empirical results on several datasets are relatively lower than the baseline results, the effectiveness of the proposed method remains unclear.**
> > > > >
> > > > > We respectfully disagree with the reviewer on this point. We note that In 1D and 2D CCNN outperforms or performs in par to state of the art models (it obtains state of the art on, e.g.,  Long-Range Arena tasks, SpeechCommands, Sequential Cifar, all of this without any structural changes. In 3D, although the performance of point-cloud is below the current state of the art, it is still better than the Perceiver --the only existing general purpose model--.
> > > > >
> > > > > Based on the previous results, and given that we use a single architecture across all these tasks, we consider the obtained accuracies to be convincing. We hope that the reviewer can see the added value of the empirical evidence of our method.
> > > > >
> > > > > **[Final words]** We hope that these responses clarify your questions and concerns. Please let us know if you have any follow-up / additional questions.
> > > > >
> > > > > Best regards,
> > > > >
> > > > > The Authors

---

> > > > > > ### Author Response · Authors · 2022-11-18
> > > > > > **Final thoughts**
> > > > > >
> > > > > > Dear Reviewer RGNG,
> > > > > >
> > > > > > As the rebuttal period is closing shortly, please let us know if you have any further questions or if we can provide further clarification!
> > > > > >
> > > > > > --The Authors

---

### Official Review · Reviewer_jCEb · 2022-10-24

**Confidence:** 4
**Correctness:** 4
**Technical Novelty And Significance:** 3
**Empirical Novelty And Significance:** 3
**Recommendation:** 3

**Clarity, Quality, Novelty And Reproducibility:**

### Clarity
This paper should clearly state its technical contribution.
### Quality
The writing quality is fair.

### Novelty
The novelty is limited. The core idea of the proposed CCNN is the continuous kernel convolution, however, it has been explored and proposed by previous publications. The proposed CCNN adopts the continuous kernels to adapt for different shapes of input, which is below the threshold of technical novelty.

### Reproducibility
The authors provide some implementation details of the experiments but it’s still hard for reproducing the methods and experimental results.


**Strength And Weaknesses:**

### Strength
1. This paper presents a data-independent operation, i.e., continuous convolution operator, which is adaptive to the shape, resolution, and dimension. The continuous convolution based on Continuous Kernels maps the spatial (relative) locations to the parameters.
2. This paper builds a continuous convolutional networks based on the continuous kernel and depth-wise continuous convolutions, namely CCNN, and provides a series of optimization strategies, including the initialization, regularization, and etc.
3. This paper adopts the same architecture on several tasks with different input resolutions or shapes, e.g., 2D images and 3D point clouds.
4. Experimental results of this paper on varieties of benchmarks of various input shapes are good.

### Weakness
1. The novelty of this paper is limited. The proposed continuous convolution has been explored in the most relevant papers [1,2], which share the same motivation and similar approach with this submission. The authors SHOULD clearly state the contribution of this paper and the differences compared to others’ works. The technical novelty of the proposed CCNN based on Continuous Kernels and FlexConv is limited for me.
2. This paper lacks detailed illustration and description about the proposed architecture, which is the main contribution in my opinion since the continuous convolution can not be regarded as a novel contribution of this paper.
3.  I don’t agree that data-dependent architectures are terrible or limited by shapes, resolutions, or dimensions. Firstly, data-dependent architectures bring more inducive bias for specific tasks, which has better characteristics compared to the so-called universal architectures. Secondly, data-dependent or hand-crafted architectures can perform better on specific data/tasks in terms of both speed and accuracy, and most practical applications are driven by data-dependent architectures. Moreover, we now have massive versatile transformers and neural architecture search mechanisms, and I’m concerned about the predominance of the proposed CCNN.
4. This paper lacks the exact inference speed, training speed, or latency on the given devices, e.g., NVIDIA GPU. It matters for me.
5. In Tab.1, the authors should provide more comparisons with the newer methods, e.g., vision transformers. I’m glad to see that the proposed CCNN can perform better with less or comparable computation budget when compared to recent vision transformers.
6. I’m concerned about the transferring ability on 2D inputs, i.e., 2D images. For example, training the proposed CCNN and a normal CNN which have similar hierarchical architecture and parameters on ImageNet or other datasets and evaluate the performance on lower-resolution datasets. Providing fair comparisons and experimental evaluations will be more convincing in my opinion.

[1] Remero et.al. CKConv: Continuous Kernel Convolution for Sequential Data. ICLR 2022.
[2] Remero et.al. FlexConv: Continuous Kernel Convolutions with Differentiable Kernel Sizes. ICLR 2022.

### Typos
Missing reference in Sec. 3.3: “with 9 parameters (Sec. ??)”.


**Summary Of The Paper:**

This paper presents Continuous Convolutional Neural Networks, which can process the input of arbitrary resolution, length, and dimensionality. Compared to conventional convolutional neural networks which adopts fixed-size kernels, the proposed continuous convolutional networks adopt dynamic and continuous kernels, which removes the dependencies of the input data. Based on the presented continuous convolution operation, this paper designs a deep continuous convolutional network and apply it to several tasks with different input shapes and resolutions.

**Summary Of The Review:**

Considering the limited technical novelty and contribution of this paper, I think this paper is not qualified as a conference paper. This paper is highly coincident with previous works.

---

> ### Author Response · Authors · 2022-11-14
> **First response -- reviewer jCEb**
>
> Dear reviewer jCEb,
>
> First of all, we would like to thank you very much for your thorough review. We sincerely appreciate the time you spent in evaluating our work, and very much appreciate your comments.
>
> Here we will answer your questions, comments and concerns:
>
> **The novelty of this paper is limited. The proposed continuous convolution has been explored in the most relevant papers [1,2], which share the same motivation and similar approach with this submission. The authors SHOULD clearly state the contribution of this paper and the differences compared to others’ works. The technical novelty of the proposed CCNN based on Continuous Kernels and FlexConv is limited for me.**
>
> We have realised that our contribution was not entirely clear in the current form of the paper (as pointed by multiple reviewers). To solve this issue, we have (i) changed the title, (ii) added a contributions subsection to the introduction and (iii) restructured Sec. 4. We hope that the current form of our paper highlights our contributions more clearly and properly sets our work apart from previous work.
>
> The reviewer raises concerns about the technical novelty of the work. We would like to emphasise that the focus of this work contrasts with previous publications. Whereas this work aims for a general purpose CCNN architecture, previous works on continuous kernels have a different focus, e.g. operating on point-cloud data [1, 2, 3], modelling of large sequence data [4] or learning adequate receptive fields [5].  We identify aspects of previous works that make them unfit for architecture unification, and subsequently provide the required modifications to yield a framework that can be deployed across wildly different modalities without structural changes. In this regard, we feel our work contributes (i) insight into those aspects of current CNN formulations that make them task-specific, (ii) a unified framework that reliefs architectures from these considerations, and (iii) an empirical verification of this framework several different data modalities.
>
> In addition, we would like to point out that other approaches with a similar aim, remarkably the Perceiver [6], also build on top of existing methods and do not propose much changes upon these works. Yet, the vision and aim of the paper –which is the main contribution of the Perceiver– sets it apart from previous work, and has made it a very relevant paper to the community.
>
> [1] Schütt, K. T., Sauceda, H. E., Kindermans, P. J., Tkatchenko, A., & Müller, K. R. (2018). Schnet–a deep learning architecture for molecules and materials. The Journal of Chemical Physics, 148(24), 241722.
>
> [2] Wu, W., Qi, Z., & Fuxin, L. (2019). Pointconv: Deep convolutional networks on 3d point clouds. In Proceedings of the IEEE/CVF Conference on Computer Vision and Pattern Recognition (pp. 9621-9630).
>
> [3] Jia, X., De Brabandere, B., Tuytelaars, T., & Gool, L. V. (2016). Dynamic filter networks. Advances in neural information processing systems, 29.
>
> [4] Romero, D. W., Kuzina, A., Bekkers, E. J., Tomczak, J. M., & Hoogendoorn, M. (2021). Ckconv: Continuous kernel convolution for sequential data. arXiv preprint arXiv:2102.02611.
>
> [5] Romero, D. W., Bruintjes, R. J., Tomczak, J. M., Bekkers, E. J., Hoogendoorn, M., & van Gemert, J. C. (2021). Flexconv: Continuous kernel convolutions with differentiable kernel sizes. arXiv preprint arXiv:2110.08059.
>
> [6] Jaegle A, Gimeno F, Brock A, Vinyals O, Zisserman A, Carreira J. Perceiver: General perception with iterative attention. InInternational conference on machine learning 2021 Jul 1 (pp. 4651-4664). PMLR.
>
> **This paper lacks detailed illustration and description about the proposed architecture, which is the main contribution in my opinion since the continuous convolution can not be regarded as a novel contribution of this paper.**
>
> We agree with the reviewer. We have now extensively modified our paper to illustrate the proposed architecture as well as the proposed changes. In addition, we have added multiple figures and experiments to illustrate the importance of each of these modifications (See Figures across the paper as well as Sec D.2).

---

> > ### Author Response · Authors · 2022-11-14
> > **First response -- reviewer jCEb -- continuation**
> >
> > **I don’t agree that data-dependent architectures are terrible or limited by shapes, resolutions, or dimensions. Firstly, data-dependent architectures bring more inductive bias for specific tasks, which has better characteristics compared to the so-called universal architectures.**
> >
> > We would like to emphasise that we do not oppose employing inductive biases in network design. On the contrary, this is why we use convolutions and not other types of universal architectures, e.g.Transformers, MLPs. Using convolutions make our networks translation equivariant –as other CNNs–, and thus respect their inductive biases. In addition, we would like to point out that our model actually includes an additional inductive bias compared to conventional CNNs by requiring the convolutional kernel to be continuous, which is a desirable property as it reflects the continuous nature of Euclidean data (audio, images, point-clouds).
> >
> > The contribution of our paper is that it decouples architecture design from parameters that do not reflect any task-specific inductive bias (kernel size, pooling, depth), but that rather are necessary to account for the fact that conventional CNNs do not respect the continuous nature of the data, and thus are unable to model long term dependencies at every layer. Note that many differences between task-specific CNN architectures are not a result of the inclusion of inductive biases into model design, but rather an artefact of the discrete nature of convolutional kernels used in them. For example, the need for more pooling operations in architectures applied to higher resolution images does not reflect any specific inductive bias, but it is simply included to obtain an adequate increase in receptive field in order to model dependencies over the increased spatial resolution of the input with discrete convolutional kernels.
> >
> > We propose to move to a paradigm which removes the need for these arbitrary choices in architectural hyperparameters; a higher resolution input image should not be reflected by architectural modifications, as the network should model the same underlying function only in higher fidelity.
> >
> > In summary, the proposed unified framework improves transferability of insights in network architecture design and reduces the need for task-specific network modifications that do not reflect valuable inductive biases. The value of our work lies in proposing a unification for conventional CNNs which are normally separated by seemingly arbitrary hyperparameter choices (e.g. kernel size / resolution, network depth, number of pooling operations). This viewpoint leads to a simplified architectural design without performance loss, which we validate by obtaining state-of-the-art performance on multiple datasets.
> >
> > **Secondly, data-dependent or hand-crafted architectures can perform better on specific data/tasks in terms of both speed and accuracy, and most practical applications are driven by data-dependent architectures. Moreover, we now have massive versatile transformers and neural architecture search mechanisms, and I’m concerned about the predominance of the proposed CCNN.**
> >
> > First, we would like to highlight that it is not true that task-specific architectures dominate exclusively in our experiments. The proposed unified architecture either outperforms or is on par with existing task-specific works on Long-Range Arena, SpeechCommands, Sequential Cifar, Sequential MNIST and Sequential Permuted Cifar, without task specific modifications.
> >
> > Second, with regard to speed, we agree with the reviewer that task-specific architectures can be more efficient. In fact, CCNNs are the first step towards the end goal of this research direction: creating a neural network that adjusts its architecture during training, i.e., the depth, the number of channels, the downsampling layers, etc, to satisfy efficiency and accuracy objectives. We are currently working on a follow-up paper that accomplishes just that. However, to that end it is necessary to have a baseline architecture first that is able to work properly on wildly different modalities, such that it can be subsequently adapted to particular tasks during training. Funnily, this observation thights nicely to the final part of your comment :)
> >
> > Finally, please note that versatile transformers are still problematic because, as you mentioned, they do not respect the inductive biases of Euclidean data. Consequently, we believe expect CCNNs might have large impact in the future.

---

> > > ### Author Response · Authors · 2022-11-14
> > > **First response -- reviewer jCEb -- continuation 2**
> > >
> > > **This paper lacks the exact inference speed, training speed, or latency on the given devices, e.g., NVIDIA GPU. It matters for me.**
> > >
> > > We agree with the reviewer that inference speed is an important consideration in model comparison. To add further transparency to our manuscript, we added information on the computational complexity of our method to Appx. D. We show computational overhead in training speed compared to discrete CNNs, but remain in similar order of magnitude. Moreover, we compare the layers in our network against the efficient S4 layer [1] (see Appx. D.1) and show similar performance in terms of speed (although CCNN is marginally faster for input lengths up to 8192). Our use of convolutional layers through multiplication in the Fourier domain does lead to increased memory usage, but differences remain a constant factor. Compared to LSSL [2], we show to be asymptotically more efficient both in terms of speed and memory usage. We would also like to note that our code is already publicly available. However, we have removed the link from the submission to respect the double-blind review process.
> > >
> > > [1] Efficiently modeling long sequences with structured state spaces. ICLR2022
> > > [2] Gu, A., Johnson, I., Goel, K., Saab, K., Dao, T., Rudra, A., & Ré, C. (2021). Combining recurrent, convolutional, and continuous-time models with linear state space layers. Advances in neural information processing systems, 34, 572-585.
> > >
> > > **In Tab.1, the authors should provide more comparisons with the newer methods, e.g., vision transformers. I’m glad to see that the proposed CCNN can perform better with less or comparable computation budget when compared to recent vision transformers.**
> > >
> > > We recognize the value of comparison with more recent Vision Transformer architectures, and as such, we have added results for two ViT architectures to Tab. 1 (the original ViT and the Swin Transformer). We note that, due to the low inductive biases of ViT, it performs very badly on CIFAR10 and CIFAR100 if not pretrained on very large datasets. This is addressed in the Swin transformer (which adds convolutions in an intricate form to alleviate this issue).
> > >
> > > First, note that CCNN is competitive with the Swin transformer, and outperforms ViT. Second, we highlight how specific the architecture of the Swin Transformer must be made for it to work properly. CCNNs on the other hand is able to work well on very different modalities –not only 2D– without any structural changes.
> > >
> > > **I’m concerned about the transferring ability on 2D inputs, i.e., 2D images. For example, training the proposed CCNN and a normal CNN which have similar hierarchical architecture and parameters on ImageNet or other datasets and evaluate the performance on lower-resolution datasets. Providing fair comparisons and experimental evaluations will be more convincing in my opinion.**
> > >
> > > We do have an experiment for resolution transfer of Speech Commands. We have not done this for 2D because the transferability phenomenon of CNNs in 2D has been previously shown in [1] (see Section 4.3), and in [2]. They show that conventional CNNs do not generalise properly to other resolutions but FlexConvs with MAGNets and alias-free S4 models do. We have included additional discussions on this topic and, if the reviewer prefers, we can include additional experiments with resolution changes to the manuscript. However, we do emphasise that this has been previously shown.
> > >
> > > [1] Romero, D. W., Bruintjes, R. J., Tomczak, J. M., Bekkers, E. J., Hoogendoorn, M., & van Gemert, J. C. (2021). Flexconv: Continuous kernel convolutions with differentiable kernel sizes. arXiv preprint arXiv:2110.08059.
> > >
> > > [2] Nguyen E, Goel K, Gu A, Downs GW, Shah P, Dao T, Baccus SA, Ré C. S4ND: Modeling Images and Videos as Multidimensional Signals Using State Spaces. arXiv preprint arXiv:2210.06583. 2022 Oct 12.
> > >
> > > **Reproducibility.**
> > >
> > > The reviewer raises concerns regarding the reproducibility of our manuscript. To address this, we have now provided an anonymized version of the codebase for this paper as supplementary material. Moreover, we would like to note that our code is already publicly available, but we have removed the link from the submission in order to respect the double-blind review process.
> > >
> > > **[Final words]** We hope that these responses clarify your questions and concerns. Please let us know if you have any follow-up / additional questions.
> > >
> > > Best regards,
> > >
> > > The Authors

---

> > > > ### Author Response · Authors · 2022-11-18
> > > > **Final thoughts**
> > > >
> > > > Dear Reviewer jCEb,
> > > >
> > > > As the rebuttal period is closing shortly, please let us know if you have any further questions or if we can provide further clarification!
> > > >
> > > > --The Authors

---

> > > > > ### Author Response · Authors · 2022-12-04
> > > > > **Are your concerns addressed? Rebuttal period closing.**
> > > > >
> > > > > Dear Reviewer jCEb,
> > > > >
> > > > > Thank you for evaluating our work.
> > > > >
> > > > > We would kindly like to ask whether our rebuttal addressed your questions and concerns. Since the rebuttal time is coming to an end, we won't be able to respond much longer if any concern remains unanswered.
> > > > >
> > > > > We are very interested in knowing your opinion and thoughts and we would deeply appreciate if you could share them with us.
> > > > >
> > > > > Thank you in advance!
> > > > >
> > > > > Best regards,
> > > > >
> > > > > The authors

---

### Official Review · Reviewer_KStu · 2022-10-24

**Confidence:** 3
**Correctness:** 3
**Technical Novelty And Significance:** 3
**Empirical Novelty And Significance:** 3
**Recommendation:** 6

**Clarity, Quality, Novelty And Reproducibility:**

The paper is well organized. The proposed Continuous CNN is new to me. The implementation details are well provided but the code is not available.

**Strength And Weaknesses:**

Strength:
- The motivation to construct a unified convolution neural network is interesting.
- The Continuous CNN is a reasonable solution for general architecture.
- Experimental results on a range of sequence, image and point-cloud datasets show the effectiveness of the proposed method.

Weaknesses:
- What about the computational complexity? I'm afraid it will be large since there is no downsampling.
- The number of parameters will be large since the method introduces a kernel neural network.
- To make neural network independently from the input length, or resolution, there are several other alternatives like deformable convolution [1] and graph convolution [2]. Why not using these methods? Please include discussion.
- Just a suggestion and not necessary: could the proposed CCNN trained once on all the datasets in Table 1 and work well on all of them? Large-scale ImageNet performance?
- Typos: "??" in footnote in page 4.
- The writting of the paper should be improved: what are $*$ and $\tilde{x}$ in Eq.(1).

[1] Dai J, Qi H, Xiong Y, et al. Deformable convolutional networks[C]//Proceedings of the IEEE international conference on computer vision. 2017: 764-773.

[2] Han K, Wang Y, Guo J, et al. Vision GNN: An Image is Worth Graph of Nodes[J]. arXiv preprint arXiv:2206.00272, 2022.

**Summary Of The Paper:**

The paper aims to make CNN suitable for data of arbitrary resolution, dimensionality and length without any structural changes. A Continuous CNN is proposed by introducing continuous convolutional kernels which is a data independent parameterization for convolutional weight. The proposed CCNN can work on on sequential (1D), visual (2D) and point-cloud (3D) tasks with the same architecture.

**Summary Of The Review:**

I like the idea to unify CNN architectures on various tasks.

---

> ### Author Response · Authors · 2022-11-14
> **First response -- reviewer KStu**
>
> Dear reviewer KStu,
>
> First of all, we would like to thank you very much for your thorough review. We sincerely appreciate the time you spent in evaluating our work, and very much appreciate your comments.
>
> Here we will answer your questions, comments and concerns:
>
> **The writting of the paper should be improved.**
>
> Thank you for the suggestion. We have extensively improved the readability and structure of the paper. Any remaining remarks are highly appreciated!
>
> **What about the computational complexity? I'm afraid it will be large since there is no downsampling.**
>
> It is true that removing downsampling can have repercussions regarding the computational complexity of the model. However, as we model long range dependencies at each layer, our networks are very shallow --CCNN_{4, 140} and CCNN_{6, 380} have 5 and 7 convolutional layers in total--, and we can rely on FFT convolutions to strongly reduce the computational complexity of convolutions with large kernels. In addition, we would like to point out that the only existing general purpose architecture, the Perceiver [1], relies on self-attention and scales significantly worse than our convolutional approach. Hence, CCNNs do scale gracefully in comparison to existing alternatives (we have added a discussion wrt the Perceiver in the Related Work section).
>
> With that being said, it is true that computational complexity remains an issue, certainly for very large data. We have updated our manuscript to reflect this, and have pointed out some possible research directions that could be used to further reduce the computational complexity, e.g., via learnable downsampling [2]. To add further transparency to our manuscript, we added an extensive experimental evaluation of the computational complexity of our method to the appendix and have linked this section to the main text.
>
> [1] Jaegle A, Gimeno F, Brock A, Vinyals O, Zisserman A, Carreira J. Perceiver: General perception with iterative attention. InInternational conference on machine learning 2021 Jul 1 (pp. 4651-4664). PMLR.
>
> [2] Riad R, Teboul O, Grangier D, Zeghidour N. Learning strides in convolutional neural networks. arXiv preprint arXiv:2202.01653. 2022 Feb 3.
>
> **The number of parameters will be large since the method introduces a kernel neural network.**
>
> Fortunately, this is not the case. Note that the networks used to produce the kernels are extremely small (3 layers & 32 hidden channels). As explained in [1] – find a copy of the relevant part under this response–, this parameterization scales much better than modelling kernels as independent weights. It is in fact the parameter efficiency of continuous kernels that allows us to construct very powerful networks with a small parameter count –we achieve SOTA on several datasets with 200k and even showcase much smaller networks (down to 6k params) on point-clouds–. We have modified our manuscript to make this clearer.
>
> [1] Romero, D. W., Kuzina, A., Bekkers, E. J., Tomczak, J. M., & Hoogendoorn, M. (2021). Ckconv: Continuous kernel convolution for sequential data. arXiv preprint arXiv:2102.02611.
>
> (From Romero et al):
>
> >**Parameter-efficient large convolutional kernels.** CKConvs construct large complex kernels with a fixed parameter budget. For large input sequences, this results in large savings in the number of parameters required to construct global kernels with conventional CNNs. For sequences from the pMNIST (length = 784) and SC_raw (length = 16000) datasets, a conventional CNN with global kernels would require 2.14M and 46.68M of parameters, respectively, for a model equivalent to our CKCNN (100K). In other words, our kernel parameterization allows us to construct CKCNNs that are 21, 84 and 445, 71 times smaller than corresponding conventional CNNs for these datasets. Detailed exploration on the effect of our efficient continuous kernel parameterizations in optimization, overfitting and generalization is an interesting direction for future research.

---

> > ### Author Response · Authors · 2022-11-14
> > **First response -- reviewer KStu -- continuation**
> >
> > **To make neural network independently from the input length, or resolution, there are several other alternatives like deformable convolution [1] and graph convolution [2]. Why not using these methods? Please include discussion.**
> >
> > Indeed. Based on our analysis (Sec. 3), any neural architecture able to model long-term dependencies without input dependent pooling and depth, and whose parameter count does not change in order to consider different lengths and resolutions, can be used as a general purpose model. We discuss this in Section 3 (more clearly in the revised version).
> >
> > Unfortunately, this is not the case for deformable convolutions, as both the offsets and kernel values are modelled as discrete values. Hence they are still tied to resolutions and cannot on dense global context –a discussion between FlexConv and deformable convolutions is provided in [1]. There, it is shown that deformable convolutions are a low-frequency representation of a dense kernel, and thus, is unable to model dense long-term dependencies.
> >
> > On the other hand, other model architectures such as graph convolutions and transformers could in principle be used. However, these architectures are computationally much more expensive than convolution, and prevent the network from using Fourier convolutions –which we extensively use to model long term dependencies across large regular inputs in a computationally efficient manner.
> >
> > The previous observations outline why we prefer CKConvs to other methods. As outlined now in the Related Work, other alternatives, e.g., state-space models (Gu et al. 2022) could be used. However, these have other important limitations.
> >
> > We appreciate the valuable suggestions. We have now extended the discussion and related works section to include your observations. In addition, an in depth discussion with other possible architectures is provided in the Extended Related Work section in Appx. A.
> >
> > [1] Romero, D. W., Bruintjes, R. J., Tomczak, J. M., Bekkers, E. J., Hoogendoorn, M., & van Gemert, J. C. (2021). Flexconv: Continuous kernel convolutions with differentiable kernel sizes. arXiv preprint arXiv:2110.08059.
> >
> > [2] Efficiently modeling long sequences with structured state spaces. ICLR2022
> >
> > **Just a suggestion and not necessary: could the proposed CCNN trained once on all the datasets in Table 1 and work well on all of them?**
> >
> > This is a very interesting suggestion and definitely an interesting future research direction. A similar suggestion was posed by the reviewer Av5v with regard to multi-modal and cross-modal training. Here’s a copy of our response to him:
> >
> > >“Multi-modal or cross-modal application of our model is outside our current scope, mainly since to go to multi-modal training, we consider it important to have an unifying architecture that performs well across wildly different modalities first. However, as noted by the reviewer, the proposed method does provide a handle for weight sharing over multiple modalities through a shared latent space (for instance by sharing part of the kernel network). We thank the reviewer for this suggestion. We have stated this explicitly in our Future Work section, and will definitely pursue this line of work in the future.“
> >
> > **Typos: "??" in footnote in page 4.**
> >
> > These have been addressed.
> >
> > **[Final words]** We hope that these responses clarify your questions and concerns. Please let us know if you have any follow-up / additional questions.
> >
> > Best regards,
> >
> > The Authors

---

> > > ### Author Response · Authors · 2022-11-18
> > > **Final thoughts**
> > >
> > > Dear Reviewer KStu,
> > >
> > > As the rebuttal period is closing shortly, please let us know if you have any further questions or if we can provide further clarification!
> > >
> > > --The Authors

---

> > > > ### Comment · Reviewer_KStu · 2022-11-30
> > > > **Thanks**
> > > >
> > > > Thanks for the response. I have no more questions.

---

### Official Review · Reviewer_Av5z · 2022-10-26

**Confidence:** 4
**Correctness:** 4
**Technical Novelty And Significance:** 2
**Empirical Novelty And Significance:** 3
**Recommendation:** 6

**Clarity, Quality, Novelty And Reproducibility:**

The paper is of OK quality and clarity. The originality of the method is limited as discussed above but the reported results and discussions could be of help for a sub-field in the community.

**Strength And Weaknesses:**

Pros:
1. The proposed idea is interesting.
2. The empirical results are supporting.

Cons:
1. The technical novelty is rather limited as the building blocks are mainly from exisiting work, e.g., FlexConv, although the reviewer does recognize the contribution of all the empirical results and discussions.
2. One important aspect is missing: transformer is not only surprising as one architecture could be applied to different type of inputs. The more practical and important fact is that different inputs could indeed share most of the weights of the network [r1,r2].
a. Therefore, it will be really more exciting if the authors also explores the setting where different types of inputs/tasks share the same set of network weights.
b. Another interesting exploration is to interpret the network weights trained on different data: does model trained on different data in the same modality share certain weighs? do models trained on different types of data even also share certain patterns?

[r1] All in one: Exploring unified video-language pre-training
[r2] PolyViT: Co-training Vision Transformers on Images, Videos and Audio



Minor:
"Sec. ??" appears multiple times. For example, in Sec 3.3, in the second paragraph, "Sec. ??".

**Summary Of The Paper:**

In this paper, the authors proposed a unified CNN architecture that is tested on multiple datasets. The main idea is to learn a hyper-network to predict the discrete kernel given coordinates as input, which is instantiated based on an existing kernel-size differentiable convolution (FlexConv). Extensive experimental results are provided on multiple benchmark datastes covering inputs of different dimentionalities and modalities.

**Summary Of The Review:**

Despite of limited novelty on the technical side and some missing exploration to further support the motivation of unifying the architecture among modalities, the reviewer still feels the current results and discussions could be possibly helpful for the community.

---

> ### Author Response · Authors · 2022-11-14
> **First response -- reviewer Av5z**
>
> Dear reviewer Av5v,
>
> First of all, we would like to thank you very much for your review. We sincerely appreciate the time you spent in evaluating our work, and very much appreciate your comments.
>
> Here we will answer to your questions, comments and concerns:
>
> **The technical novelty is rather limited as the building blocks are mainly from exisiting work, e.g., FlexConv, although the reviewer does recognize the contribution of all the empirical results and discussions.**
>
> We have realised that our contribution was not entirely clear in the current form of the paper (as pointed by multiple reviewers). To solve this issue, we have (i) changed the title, (ii) added a contributions subsection to the introduction and (iii) restructured Sec. 4. We hope that the current form of our paper highlights our contributions more clearly and properly sets our work apart from previous work.
>
> We would like to emphasise that the focus of this work contrasts with previous publications. Whereas this work aims for a general purpose CCNN architecture, previous works on continuous kernels have a different focus, e.g. operating on point-cloud data [1, 2, 3], modelling of large sequence data [4] or learning adequate receptive fields [5].  We identify aspects of previous works that make them unfit for architecture unification, and subsequently provide the required modifications to yield a framework that can be deployed across wildly different modalities. In this regard, we feel our work contributes (i) insight into those aspects of current CNN formulations that make them task-specific, (ii) a unified framework that reliefs architectures from these considerations, and (iii) an empirical verification of this framework on several data modalities.
>
> In addition, we would like to point out that other approaches with a similar aim, remarkably the Perceiver [6], also build on top of existing methods and do not propose much changes upon these works. Yet, the vision and aim of the paper –which is their main contribution– sets it apart from previous work, and has made it a very relevant paper to the community.
>
> [1] Schütt, K. T., Sauceda, H. E., Kindermans, P. J., Tkatchenko, A., & Müller, K. R. (2018). Schnet–a deep learning architecture for molecules and materials. The Journal of Chemical Physics, 148(24), 241722.
>
> [2] Wu, W., Qi, Z., & Fuxin, L. (2019). Pointconv: Deep convolutional networks on 3d point clouds. In Proceedings of the IEEE/CVF Conference on Computer Vision and Pattern Recognition (pp. 9621-9630).
>
> [3] Jia, X., De Brabandere, B., Tuytelaars, T., & Gool, L. V. (2016). Dynamic filter networks. Advances in neural information processing systems, 29.
>
> [4] Romero, D. W., Kuzina, A., Bekkers, E. J., Tomczak, J. M., & Hoogendoorn, M. (2021). Ckconv: Continuous kernel convolution for sequential data. arXiv preprint arXiv:2102.02611.
>
> [5] Romero, D. W., Bruintjes, R. J., Tomczak, J. M., Bekkers, E. J., Hoogendoorn, M., & van Gemert, J. C. (2021). Flexconv: Continuous kernel convolutions with differentiable kernel sizes. arXiv preprint arXiv:2110.08059.
>
> [6] Jaegle A, Gimeno F, Brock A, Vinyals O, Zisserman A, Carreira J. Perceiver: General perception with iterative attention. InInternational conference on machine learning 2021 Jul 1 (pp. 4651-4664). PMLR.
>
> **One important aspect is missing: transformer is not only surprising as one architecture could be applied to different type of inputs.**
>
> Note that, except for the Perceiver, transformers still require encoding task-specific architectural considerations (e.g. convolutional biases allow ViT, Swin Transformer to perform on image data). While very relevant in their own right  –especially in the context of incorporating task-specific inductive biases into network architecture–, we show that architectural characteristics  (e.g. network depth, number of pooling operations) are not necessary to reflect task-specific inductive biases, and can be removed from CNN architectures.
>
> **The more practical and important fact is that different inputs could indeed share most of the weights of the network [r1,r2]. a. Therefore, it will be really more exciting if the authors also explores the setting where different types of inputs/tasks share the same set of network weights.**
>
> The reviewer proposes some very interesting ideas. Multi-modal or cross-modal application of our model is outside our current scope, mainly since to go to multi-modal training, we consider it important to have an unifying architecture that performs well across wildly different modalities first. However, as noted by the reviewer, the proposed method does provide a handle for weight sharing over multiple modalities through a shared latent space (for instance by sharing part of the kernel network). We thank the reviewer for this suggestion. We have stated this explicitly in our Future Work section, and will definitely pursue this line of work in the future.

---

> > ### Author Response · Authors · 2022-11-14
> > **First response -- reviewer Av5z -- continuation**
> >
> > **Another interesting exploration is to interpret the network weights trained on different data: does model trained on different data in the same modality share certain weighs? do models trained on different types of data even also share certain patterns?**
> >
> > This is a very interesting question. We have now added an image (Fig. 12) in which we show what networks that consider images as flattened inputs learn. Interestingly, we observe that in this setting CCNNs do their best to construct representations that resemble the 2D structure of the images. For instance, for CIFAR10, we see a clear periodic pattern every 32 steps, corresponding to the width of the image in 2D. In other words, the network learns to model (flattened) 2D patterns in 1D.
> >
> > **Minor: "Sec. ??" appears multiple times. For example, in Sec 3.3, in the second paragraph, "Sec. ??".**
> >
> > These have been addressed.
> >
> > **[Final words]** We hope that these responses clarify your questions and concerns. Please let us know if you have any follow-up / additional questions.
> >
> > Best regards,
> >
> > The Authors

---

> > > ### Author Response · Authors · 2022-11-18
> > > **Final thoughts**
> > >
> > > Dear Reviewer Av5v,
> > >
> > > As the rebuttal period is closing shortly, please let us know if you have any further questions or if we can provide further clarification!
> > >
> > > --The Authors

---

> ### Author Response · Authors · 2022-12-04
> **Are your concerns addressed? Rebuttal period closing.**
>
> Dear Reviewer  Av5z,
>
> Thank you for evaluating our work.
>
> We would kindly like to ask whether our rebuttal addressed your questions and concerns. Since the rebuttal time is coming to an end, we won't be able to respond much longer if any concern remains unanswered.
>
> We are very interested in knowing your opinion and thoughts and we would deeply appreciate if you could share them with us.
>
> Thank you in advance!
>
> Best regards,
>
> The authors

---

### Author Response · Authors · 2022-11-18
**General response -- overview of changes --**

We thank all reviewers for their thorough investigations of our work, and for investing the time to write out valuable insights and criticisms.

We feel that our manuscript greatly improved by incorporating clarifications with regard to the concerns that were raised. Most importantly, multiple reviewers raised concerns regarding the contributions of our work. We recognize that our contribution was not entirely clear in the previous form of the paper. To solve this issue, we have (i) changed the title, (ii) added a contributions subsection to the introduction and (iii) restructured Sec. 4. We hope the current form of the manuscript highlights our contributions more clearly and properly sets our work apart from previous work.

Furthermore, we have included an extensive list of ablations that encompass network components and computational complexity.

In addition, we have now included an anonymized version of our codebase as supplementary material to improve reproducibility of our work.

Please let us know if you have any follow-up / additional remarks. We are happy to respond them.

Best regards,

The Authors

---

### Decision · Program_Chairs · 2023-01-20

**Decision:**

Accept: poster

**Justification For Why Not Higher Score:**

Novelty concerns

**Justification For Why Not Lower Score:**

Strong and diverse experimental results.

**Metareview: Summary, Strengths And Weaknesses:**

Paper Summary:
Authors present a variation of a method to parametrize convolution operations for CNNs that make the network agnostic to input dimensions and resolutions.  This is an extension of CKConv/FlexConv, which was not initially well communicated but has since been addressed in the revised version. The specific changes proposed in this work were 1) improving the initialization of the parameterization network so that the outputs have proper variance, 2) working with separable depth-wise convolutions followed by pointwise linear layers, 3) improvements to the residual blocks, and 4) adding weight decay to the parameterization networks. Experimental results demonstrate improvements to performance across a variety of tasks.


Review Summary:

Pros:
- A data independent approach to model CNNs is interesting (jCEb,Av5z,KStu)
- Applicable to several tasks of different dimensions (jCEb)
- Experimental results are good (jCEb,KStu)

Cons:
- Novelty is limited / Similar works exist (jCEb, Av5z).  -- Authors have acknowledged lack of clarity over how they built on top of prior similar works and have since revised the paper. Reviewer jCEb did not respond to changes, but AC and other reviewers lean towards acceptance based on value of experiments and modifications.
- Lacks data on inference speed and computational complexity (jCEb, Kstu, RGNG) -- Authors have suppled in Appendix D
- Concerns about transferability across resolutions (jCEb) -- Authors supplied experiments in speech and referred to prior publications that supply data requested.
- Should include experiments across different input types (Av5z). -- Authors have added a discussion about future work.
- What about other methods such as graph convolution or deformable convolution (Kstu) -- Authors added a discussion about the advantages of the proposed approach over these other approaches.
- Disagreement on statements in relation to CNN's dependence on data parameters (RGNG, jCEb) -- Authors have clarified statements.


AC Recommendation: Accept. Reviewers brought up several concerns that authors made significant efforts to address, leaving the majority of reviewers leaning to accept.

**Note From Pc:**

if the above contains the word "oral" or "spotlight" please see: "oral" presentation means -> notable-top-5% and "spotlight" means -> notable-top-25%. As stated in our emails, we are disassociating presentation type from AC recommendations